# Divergent metabolism between *Trypanosoma congolense* and *Trypanosoma brucei* results in differential sensitivity to metabolic inhibition

**Pieter C. Steketee**[1]*, **Emily A. Dickie**[2], **James Iremonger**[1], **Kathryn Crouch**[2], **Edith Paxton**[1], **Siddharth Jayaraman**[1], **Omar A. Alfituri**[1], **Georgina Awuah-Mensah**[3], **Ryan Ritchie**[2], **Achim Schnaufer**[4], **Tim Rowan**[5], **Harry P. de Koning**[6], **Catarina Gadelha**[3], **Bill Wickstead**[3], **Michael P. Barrett**[2,7], **Liam J. Morrison**[1]

1 The Roslin Institute, Royal (Dick) School of Veterinary Studies, University of Edinburgh, Edinburgh, United Kingdom, 2 Wellcome Centre for Integrative Parasitology, Institute of Infection, Immunity and Inflammation, University of Glasgow, Glasgow, United Kingdom, 3 School of Life Sciences, University of Nottingham, Nottingham, United Kingdom, 4 Institute of Immunology and Infection Research, University of Edinburgh, Edinburgh, United Kingdom, 5 Global Alliance for Livestock Veterinary Medicines, Edinburgh, United Kingdom, 6 Institute of Infection, Immunity and Inflammation, University of Glasgow, Glasgow, United Kingdom, 7 Glasgow Polyomics, University of Glasgow, Glasgow, United Kingdom

* Pieter.Steketee@ed.ac.uk

**Data Availability Statement:** RNA-seq data is deposited at GEO (accession number: GSE165290; URL: https://www.ncbi.nlm.nih.gov/geo/query/acc.

## Abstract

Animal African Trypanosomiasis (AAT) is a debilitating livestock disease prevalent across sub-Saharan Africa, a main cause of which is the protozoan parasite *Trypanosoma congolense*. In comparison to the well-studied *T. brucei*, there is a major paucity of knowledge regarding the biology of *T. congolense*. Here, we use a combination of omics technologies and novel genetic tools to characterise core metabolism in *T. congolense* mammalian-infective bloodstream-form parasites, and test whether metabolic differences compared to *T. brucei* impact upon sensitivity to metabolic inhibition. Like the bloodstream stage of *T. brucei*, glycolysis plays a major part in *T. congolense* energy metabolism. However, the rate of glucose uptake is significantly lower in bloodstream stage *T. congolense*, with cells remaining viable when cultured in concentrations as low as 2 mM. Instead of pyruvate, the primary glycolytic endpoints are succinate, malate and acetate. Transcriptomics analysis showed higher levels of transcripts associated with the mitochondrial pyruvate dehydrogenase complex, acetate generation, and the glycosomal succinate shunt in *T. congolense*, compared to *T. brucei*. Stable-isotope labelling of glucose enabled the comparison of carbon usage between *T. brucei* and *T. congolense*, highlighting differences in nucleotide and saturated fatty acid metabolism. To validate the metabolic similarities and differences, both species were treated with metabolic inhibitors, confirming that electron transport chain activity is not essential in *T. congolense*. However, the parasite exhibits increased sensitivity to inhibition of mitochondrial pyruvate import, compared to *T. brucei*. Strikingly, *T. congolense* exhibited significant resistance to inhibitors of fatty acid synthesis, including a 780-fold higher $EC_{50}$ for the lipase and fatty acid synthase inhibitor Orlistat, compared to *T. brucei*. These data highlight that bloodstream form *T. congolense* diverges from *T. brucei* in key areas of metabolism, with several features that are intermediate between bloodstream- and insect-stage *T.*

cgi?acc=GSE165290). Metabolomics data is deposited at MetaboLights (accession number: MTBLS2372; URL: www.ebi.ac.uk/metabolights/MTBLS2372). All other relevant data are within the manuscript and its Supporting Information files.

**Funding:** LM, PS, EP, HdK and MPB were funded by BBSRC grants BB/N007999/1 & BB/N007492/1, and LM, PS, EP, RR and MPB by BB/S001034/1 & BB/S00243X/1. Work by BW and CG was supported by University of Nottingham/Wellcome Trust Institutional Strategic Support Fund awards (204843/Z/16/Z), and GAM and CG by a Sir Halley Stewart Medical Research Grant (R410). MPB is funded by a Wellcome Trust core grant to the Wellcome Centre for Integrative Parasitology (grant 104111/Z/14/Z). This article is based on research funded in part by the Bill & Melinda Gates Foundation (grants OPP49064 and OPP1093639) and the UK Government Foreign, Commonwealth and Development Office (grant FCDO 203188) through GALVmed. The findings and conclusions contained within are those of the authors and do not necessarily reflect positions or policies of the Bill & Melinda Gates Foundation or the UK Government. The Roslin Institute is core funded by the BBSRC (BS/E/D/20002173). The funders had no role in study design, data collection and analysis, decision to publish, or preparation of the manuscript.

**Competing interests:** The authors have declared that no competing interests exist.

*brucei.* These results have implications for drug development, mechanisms of drug resistance and host-pathogen interactions.

## Author summary

Animal African Trypanosomiasis (AAT), also known as Nagana, is a devastating disease affecting livestock across sub-Saharan Africa. AAT is primarily caused by the parasite *Trypanosoma congolense*, yet our biological knowledge about this pathogen is poor, especially compared to the related species *T. brucei*, subspecies of which cause the human disease Sleeping Sickness. Understanding the core metabolism of *T. congolense* is crucial in order to gain insights into the infection biology of this important pathogen, as well as providing the potential to identify new drug targets. In this work, we addressed the lack of knowledge concerning *T. congolense* by carrying out a comprehensive analysis of core metabolism, and comparing the data to *T. brucei*. We then used the findings of metabolic differences to predict differential sensitivity to inhibitors of metabolic function. We show that unlike *T. brucei*, where glucose metabolism leads to high levels of pyruvate excretion, *T. congolense* metabolises glucose to other end-products, namely succinate, malate and acetate. Moreover, there are pronounced differences in the way *T. congolense* uses glucose to feed into other areas of metabolism. Further analysis also suggests that *T. congolense* mostly scavenges lipids and fatty acids, rather than synthesising them *de novo*. To validate these findings, we confirm that *T. congolense* is differentially susceptible to metabolic inhibitors compared to *T. brucei*, and that, in particular, *T. congolense* is significantly less sensitive to inhibitors of fatty acid synthesis. Our study provides a foundation of functional metabolic knowledge on *T. congolense*, with insights into how this parasite fundamentally differs from *T. brucei*.

## Introduction

The hemoflagellate protozoan parasite *Trypanosoma congolense* is a primary causative agent of animal African trypanosomiasis (AAT), which can also be caused by *T. vivax* and *T. brucei* [1]. AAT is one of the most important livestock diseases across sub-Saharan Africa and accounts for livestock deaths in excess of 3 million annually, with up to 120 million cattle at risk [2–4]. Current methods of disease control centre around chemotherapy and prophylaxis (reviewed in [3]), but the very few available veterinary trypanocidal drugs have been used extensively for decades, resulting in resistance and inadequate protection [5–8]. In contrast to *T. brucei* [9], the resistance mechanisms of *T. congolense* are still poorly understood [10]. As such, there is a critical need for the development of new and improved chemotherapeutics to manage AAT [3,11], and furthering our knowledge of how *T. congolense* develops resistance to drugs can facilitate optimising the lifetime of both existing and new drugs.

Most of our biological understanding of African trypanosomes derives from studies on *T. brucei*, subspecies of which (*T. b. gambiense* and *T. b. rhodesiense*) cause Human African Trypanosomiasis (HAT) [12]. The ability to culture both procyclic (PCF; tsetse fly) and bloodstream (BSF; mammalian) forms of *T. brucei in vitro*, combined with its tractability with respect to genetic manipulation, have enabled extensive study of this species at a molecular level [13,14]. In stark contrast, very few *T. congolense* strains are amenable to continuous BSF culture, with a single strain (IL3000) used in most studies [15]. Whilst genetic modification

has been feasible in PCF stage *T. congolense* for some time, routine BSF transfection has only recently become possible [16–18]. Additionally, although *T. congolense* exhibits a superficially similar morphology and life cycle to *T. brucei* [19,20], emerging evidence increasingly suggests that *T. brucei*, *T. congolense* and *T. vivax* exhibit some profound differences at the genomic level [21–25], including in genes and phenotypes of direct relevance to metabolism, infection biology and disease epidemiology. For example, *T. brucei* has an elaborate nucleoside transporter lineage, whilst *T. congolense* has instead diversified its nucleobase transporter lineage [21]. There is a lack of understanding to what extent these genetic differences translate into biological differences, including with respect to metabolism.

Understanding metabolism is critical to identifying how pathogens survive and thrive in the varying host environments they encounter, as well as being a means of identifying drug targets, elucidating modes of drug action and mechanisms of drug resistance [26–28]. *T. brucei* metabolism has been extensively studied, aided by the application of technologies such as liquid chromatography-mass spectrometry (LC-MS) and nuclear magnetic resonance (NMR) spectroscopy (reviewed in detail by [29,30]), which enable global profiling of the cellular metabolome.

The BSF stage of *T. brucei* utilizes the high levels of glucose that are available in the mammalian bloodstream, and depends almost exclusively on the glycolytic pathway to generate ATP [31]. The first seven steps of glycolysis are enclosed in a specialized organelle, the glycosome, which maintains its own ATP/ADP and NAD/NADH balance, allowing glycolysis to proceed at an extraordinarily high rate in comparison to most other eukaryotic cells [32]. The endpoint of glycolysis, pyruvate, is a waste product of *T. brucei*, and the majority is excreted from the cell in large quantities. However, small amounts of pyruvate are further metabolized in the mitochondrion to acetate by pyruvate dehydrogenase (PDH) and acetate:succinate CoA transferase (ASCT), a secondary, yet essential pathway [33]. The acetate generated from this pathway is utilized, at least partially, for the *de novo* synthesis of fatty acids [34]. Indeed, both BSF and PCF *T. brucei* are highly sensitive to the lipase and fatty acid synthase inhibitor Orlistat [35].

Conversely, in the absence of blood meals, glucose is scarce in the tsetse fly midgut [36], and the main energy source of PCF *T. brucei* is L-proline. Its catabolism leads to production of acetate, succinate and L-alanine in PCF, which have a more developed and active mitochondrion than BSF (including an active respiratory chain capable of generating ATP, which is inactive in BSF *T. brucei* [37]). Until recently, it was thought that PCF *T. brucei* did not exhibit an active citric acid (TCA) cycle, although recent data have shown that TCA intermediates such as succinate and 2-oxoglutarate can stimulate PCF *T. brucei* growth [38–40].

Among the glycolytic enzymes, *T. brucei* expresses three isoforms of phosphoglycerate kinase (PGK), which catalyze the conversion of 1,3-bisphosphoglycerate to 3-phosphoglycerate [41]. These are developmentally regulated, with the major isoform in BSF parasites present in the glycosome (PGK-C), whilst the primary PCF isoform is found in the cytosol (PGK-B) [42]. The localization of PGK-B in the PCF cytosol is thought to result in an ATP/ADP imbalance in the glycosome, which is rectified by upregulating the glycosomal "succinate shunt", a pathway that includes the ATP-generating phospho*enol*pyruvate carboxykinase (PEPCK)- and pyruvate phosphate dikinase (PPDK)-mediated conversion of phospho*enol*pyruvate (PEP) to oxaloacetate and pyruvate respectively [42,43]. Succinate shunt activity, combined with amino acid metabolism, results in the excretion of high levels of succinate in PCF *T. brucei* [44].

Stable isotope labelling data has revealed that BSF *T. brucei* utilize D-glucose to a greater extent than first realized, with its carbons disseminating into amino acid, lipid and nucleotide metabolism [45]. This study also showed that some of the succinate and malate excreted from BSF *T. brucei* originates from glycolysis and that, unexpectedly, inhibition of PEPCK is lethal

for this life-cycle stage [45]. It has also been shown that acetate production is essential to BSF *T. brucei*, in particular for the synthesis of fatty acids (FAs) [33]. Acetate excretion, as well as that of succinate and malate, is negligible in BSF *T. brucei* compared to that of pyruvate and L-alanine.

In contrast to *T. brucei*, the literature on metabolism in *T. congolense* is scarce. More than half a century ago it was suggested that BSF *T. congolense* has a significantly lower rate of glucose consumption compared to BSF *T. brucei* [46]. Furthermore, pyruvate is not the main glycolytic end product and instead, acetate and succinate are excreted at high levels, indicative of metabolism more akin to PCF *T. brucei* [46]. Further work has revealed additional differences that support this hypothesis [47–49]. For example, BSF *T. congolense* primarily express cytosolic PGK-B, rather than glycosomal PGK-C [49]. Transmission electron microscopy has also revealed a more developed mitochondrion in BSF *T. congolense*, with visible cristae, suggesting that mitochondrial energy metabolism could play a more prominent role in BSF *T. congolense* [50]. The high levels of acetate excretion first shown by Agosin & Von Brand [46] are consistent with this hypothesis. However, other studies have shown that BSF *T. congolense* is sensitive to inhibitors of Trypanosome Alternative Oxidase (TAO), including salicylhydroxamide (SHAM); and is insensitive to cyanide, suggesting that, as for BSF *T. brucei*, TAO is the sole terminal oxidase, responsible for reoxidising glycerol 3-phosphate, in BSF *T. congolense* [51–54]. Notably, nitroblue tetrazolium staining indicates the presence of NADH dehydrogenase (complex I) activity in BSF *T. congolense* [51]. However, to date, no studies have assessed BSF *T. congolense* sensitivity to chemical inhibition of the electron transport chain, or the $F_1F_o$-ATPase.

Post-genomic technologies allow for the generation of large datasets that enable analysis of cellular processes on a systems scale, including metabolomics and transcriptomics. Integration of these data can provide a detailed snapshot of cell metabolism at the transcript and metabolite levels and help to dissect differences between species or conditions in unprecedented detail [55]. Furthermore, this knowledge can aid in prediction and understanding of drug efficacy and mode of action.

This study aimed to generate the first comprehensive overview of the metabolome of BSF *T. congolense* IL3000 parasites, allowing a global metabolic comparison of differences between *T. congolense* and *T. brucei*. Glycolytic metabolism in BSF *T. congolense* appears to be similar to PCF *T. brucei*, particularly in terms of metabolic outputs and gene expression. However, there are pronounced differences in parasite reliance on exogenous amino acids as well as carbon dissemination into pathways involved in nucleotide and lipid metabolism, as shown by stable isotope-labelled metabolomics. Using these data, we further validated these metabolic differences in *T. congolense* by pharmacological inhibition, which highlighted increased sensitivity to inhibition of mitochondrial pyruvate uptake, as well as significant resistance to inhibition of fatty acid synthesis, tested using inhibitors of fatty acid synthase and acetyl-coA synthetase. Taken together, these results suggest that *T. congolense* and *T. brucei* differ in some fundamental aspects of their core metabolism, which may have important implications for their interactions with the mammalian host, as well as potentially impacting upon drug sensitivity.

## Results

### Comparative RNA-sequencing of *T. congolense* and *T. brucei*

To compare BSF *T. congolense* and *T. brucei* transcriptomes, RNAseq analysis was carried out on parasites cultured *in vitro* (sampled from actively dividing cultures grown to densities of $1.8–2.0 \times 10^6$ cells/mL) and trypanosome samples isolated from infected mice at first peak

parasitaemia ($10^7$ cells/mL; *ex vivo*; Fig 1). *T. congolense* (IL3000) and pleomorphic *T. brucei* (STIB 247) *in vitro* and *ex vivo* samples were prepared, in order to assess similarities and differences between trypanosomes grown in culture and those from an infection (Fig 1A and 1B), and to compare and contrast the transcriptome across the species (Fig 1C and 1D). Sequencing data were aligned to the respective genome sequences with a mean overall alignment rate of $88.8 \pm 6.5\%$ and $92.6 \pm 2.0\%$ for *T. brucei* and *T. congolense* reads, respectively. The lower alignment rate for *T. brucei* was likely due to the TREU 927 reference genome having a different VSG repertoire than STIB 247. Resultant files were sorted and filtered for quality and read counts were normalised using transcripts per million (TPM) [56]. Orthologues were inferred between the species using Orthofinder [57], in order to directly compare TPM values for 1-to-1 orthologues, as well as sum-of-TPM values for groups containing families of paralogues (e.g. hexose transporters). These normalised read counts are henceforth referred to as orthoTPM values (S1 Table). The Orthofinder dataset (S2 Table) consisted of 6,677 orthogroups (denoted with the prefix "TbTc_"), of which 5,398 (80.84%) were 1-to-1 orthologues. The Orthofinder tool was also used to predict genes only present in one of the two species (S2 Table).

There are several metabolic genes that are not present in the *T. congolense* genome, including a putative delta-4 desaturase (Tb927.10.7100), a succinate dehydrogenase subunit (SDH11; Tb927.8.6890) and mitochondrial pyruvate carrier 1 (MPC1; Tb927.9.3780). These genes are encoded in regions exhibiting high levels of synteny between the species (S1 Fig) and thus seem likely to represent genuine deletions in *T. congolense*. Furthermore, guanine deaminase (GDA; Tb927.5.4560) was not identified in the *T. congolense* genome (S2 Table). Although the GDA locus is subtelomeric in the *T. brucei* genome, and these regions are not syntenic with *T. congolense*, sequences matching GDA could not be found either in the assembled genome or the available unassembled contigs. Therefore, the most parsimonious explanation is that GDA is also not encoded in the *T. congolense* genome. Based on the redundancy in trypanosomal purine salvage pathways, it is highly unlikely that GDA would be essential in BSF *T. brucei* [58].

Differences between four sample groups were assessed based on orthoTPM values (Fig 1; full dataset in S1 Table). There was a strong intra-species correlation between the *in vitro* and *ex vivo* transcriptomes (Pearson correlation coefficient, *T. congolense* ($\rho$) = 0.914, Fig 1A; *T. brucei* $\rho$ = 0.838, Fig 1B), showing that *in vitro*-derived BSF *T. congolense* and *T. brucei* closely resemble parasites isolated from infections at the transcriptome level. Furthermore, although correlations between species in the same condition were also high (*ex vivo*: $\rho$ = 0.622, Fig 1C; *in vitro*: $\rho$ = 0.614, Fig 1D), the reduced correlation coefficient values do indicate a degree of transcriptional differences between the species.

To compare data from this study to BSF *T. congolense* transcriptomics data generated by Silvester *et al.* at ascending and peak parasitaemia [59], TPM values for each annotated *T. congolense* gene were compared directly (S2 Fig and S1 Table). There was good correlation between both *in vitro* and *ex vivo T. congolense* BSF datasets and the data from Silvester *et al.* ($\rho > 0.85$, S2 Fig), with highest correlation between the *ex vivo* and ascending data ($\rho$ = 0.905, S2 Fig), albeit the correlation between the 'ascending' and 'peak parasitaemia' in Silvester *et al.* was higher ($\rho$ = 0.988, S2 Fig).

## *T. congolense* metabolite consumption and output

Global metabolite (metabolomics) analysis of *in vitro* culture supernatant samples provides a detailed insight into the metabolic inputs and outputs of cultured cells [60]. However, high levels of medium components can often mask subtle but significant changes in culture medium composition over time. To counteract this, a modified culture medium was designed for *T.*

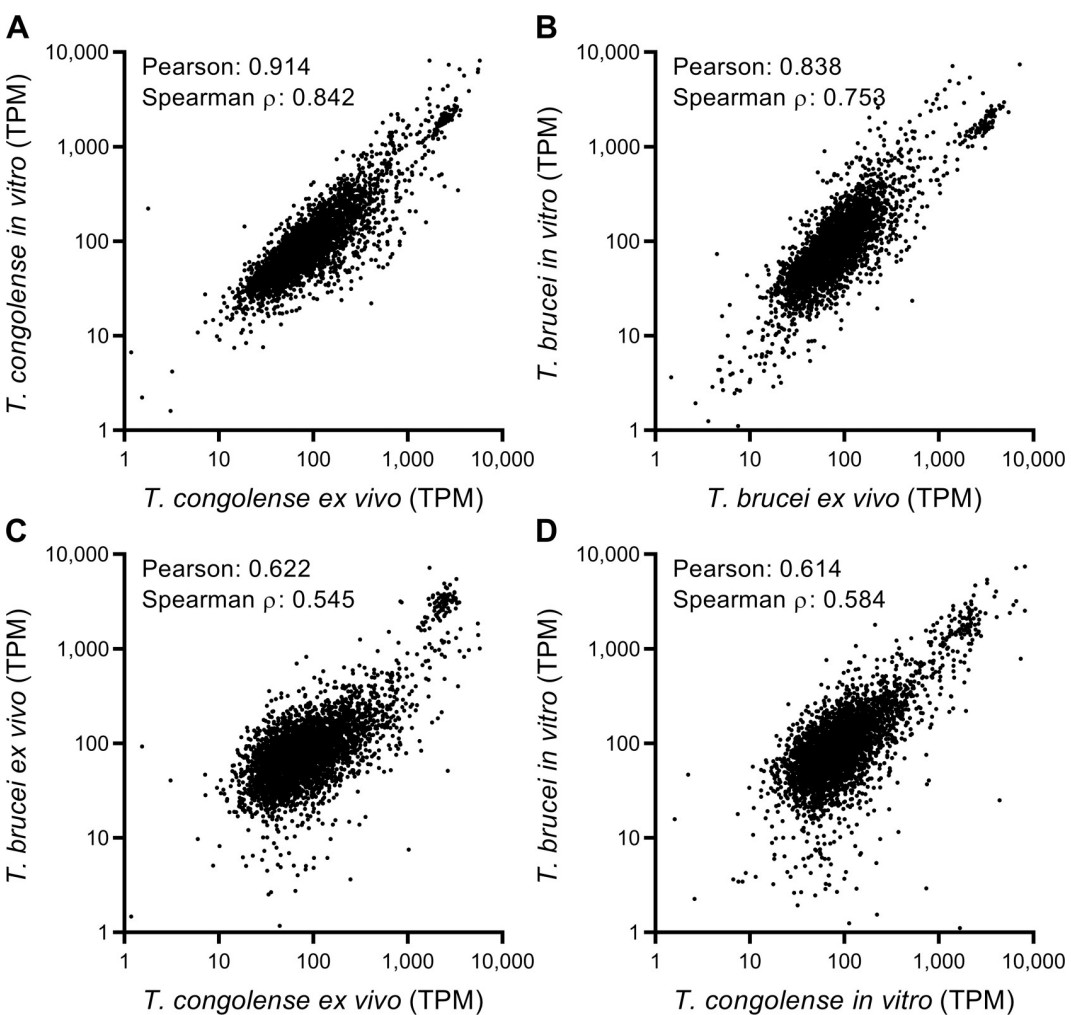

**Fig 1. Overview of comparative transcriptomics analysis of *T. brucei* and *T. congolense*, isolated from *ex vivo* and *in vitro* conditions.** RNAseq data from *T. congolense* (IL3000) and *T. brucei* (STIB 247) in both *in vitro* and *ex vivo* (from mouse infections) conditions were aligned to the species' respective reference genomes and read counts were normalised by the transcripts per million (TPM) method. To directly compare the species, a pseudogenome was generated using the Orthofinder tool [57]. TPM values from the 4 sample groups were plotted against each other to analyse correlation between conditions (A and B) and between species in the same conditions (C and D). Correlation was assessed using both Pearson correlation (Pearson's *r*) and Spearman's rank correlation coefficients.

*congolense* strain IL3000, based on previously published medium formulations (Steketee's *congolense* medium, SCM-3; for details see Materials and Methods) [17,18].

A BSF *T. congolense* IL3000 time course was initiated, with cells isolated during mid-exponential growth phase inoculated into fresh SCM-3 medium (0 h time point). Culture supernatant samples were collected at 0, 8, 24, 32, 48 and 56 hours (n = 4 at each time point) and metabolites extracted for LC-MS analysis.

A total of 290 putative metabolites were detected across all samples (207 after removing putative metabolites that did not map to metabolic pathways, e.g. peptides and medium components), of which 37 were matched to an authentic standard to confidently predict their identity (S3 Table). Of the 207 metabolites in the final dataset, 100 were putatively annotated as lipids. Annotations of putative mono- and poly-unsaturated fatty acids are of lower confidence

due to the challenges of identifying these metabolite species using this LC-MS platform (see Materials and Methods for details).

80 of the 207 putative metabolites were significantly altered across the dataset (false discovery rate-adjusted p<0.05; one-way repeated measures ANOVA; Fig 2A and S3 Table). To analyse metabolites undergoing similar changes, K-means clustering with Pearson correlation coefficient as the similarity metric was used, highlighting seven clusters with two clusters of particular interest: one containing metabolites that accumulated over time, and the other containing metabolites depleted over time (Fig 2A). $\log_2$ fold change ($\log_2$ FC) between the first and final time points (0 and 56 h, respectively) was also calculated for each metabolite (S3 Table).

Glucose, the primary energy source for *T. brucei*, whilst clearly consumed, was not fully depleted after 56 hours in *T. congolense* culture ($\log_2$ FC: -0.76; Figs 2A and 3A), in contrast to *T. brucei*, where 10 mM glucose is consumed by the same time-point [60]. Ribose, glucosamine, inosine and threonine were similarly depleted in *T. congolense* culture ($\log_2$ FC: -0.78, -0.97, -2.82 and -0.89, respectively).

In contrast, a number of metabolites accumulated in the medium (Fig 2A). The most significant of these were guanine ($\log_2$ FC: 6.34; Fig 2A), succinate ($\log_2$ FC: 5.60; Figs 2A and 3B) and (S)-malate (malate, $\log_2$ FC: 1.37; Figs 2A and 3B). Interestingly, pyruvate ($\log_2$ FC: 0.24; Fig 3B) was not excreted at the high levels relative to starting concentration, in contrast to BSF *T. brucei* culture where pyruvate secretion is consistently observed in both HMI-11 and in Creek's Minimal medium (CMM) [60]. Instead, succinate and malate appear to be the primary glycolytic outputs from BSF *T. congolense*, which is similar to PCF *T. brucei*. Elevated levels of 2-oxoglutarate and a metabolite putatively identified as 2-oxoglutaramate were observed, which potentially originate from alanine aminotransferase activity using L-glutamate and L-glutamine, respectively, as substrates [45,61]. Moreover, a significant build-up of *N*6-Acetyl-L-lysine ($\log_2$ FC: 6.30) was observed (Fig 2B). Whilst the low molecular weight of acetate means it could not be detected by the LC-MS platform used, concentrations of this molecule were measured directly using an acetate assay in samples taken at the same time points from four independent cultures, which confirmed high levels of acetate excretion by BSF *T. congolense* (Fig 3F).

Other notable observations included the depletion of several putatively identified lysophosphatidylcholine species at 56 hours (Fig 2A and S3 Table), as seen in *T. brucei* [60], coincident with increased medium levels of *sn*-glycero-3-phosphocholine, choline and choline phosphate, indicating lyso-phospholipase activity where the charged headgroup moiety of a lyso-species is cleaved from its bound fatty acid [62]. However, given the putative nature of these fatty acid annotations, we could not confidently establish the origin of the elevated choline-related metabolites. In addition, tryptophan ($\log_2$ FC: -0.74; S3 Table) was significantly consumed (p = 0.042), in contrast with cysteine ($\log_2$ FC: -0.07; p > 0.05), despite the latter being essential to *T. brucei* [63] (S3 Table).

The $\log_2$ metabolite fold changes after 56 hours of culture of *T. congolense* were compared to those of *T. brucei* grown in HMI-11 (Fig 2B) [60]. A total of 90 metabolites were identified in both datasets, with some showing divergence between the two species (Fig 2B). Several metabolites only accumulated in *T. brucei* supernatant, in particular pyruvate, D-glycerate, 2-oxoglutarate and hydroxydodecanoic acid (Fig 2B). Conversely, succinate, *N*6-acetyl-L-lysine, 4-hydroxy-4-methylglutamate, *N*6,*N*6,*N*6-trimethyl-L-lysine and choline only accumulated in *T. congolense* supernatant (Fig 2B). Whilst cystine (Fig 2B; 12) was depleted in *T. brucei* samples, this metabolite remained unchanged in those from *T. congolense*.

In summary, whilst core elements of metabolism have been conserved between BSF *T. congolense* and *T. brucei*, several pronounced differences in *T. congolense* metabolism were

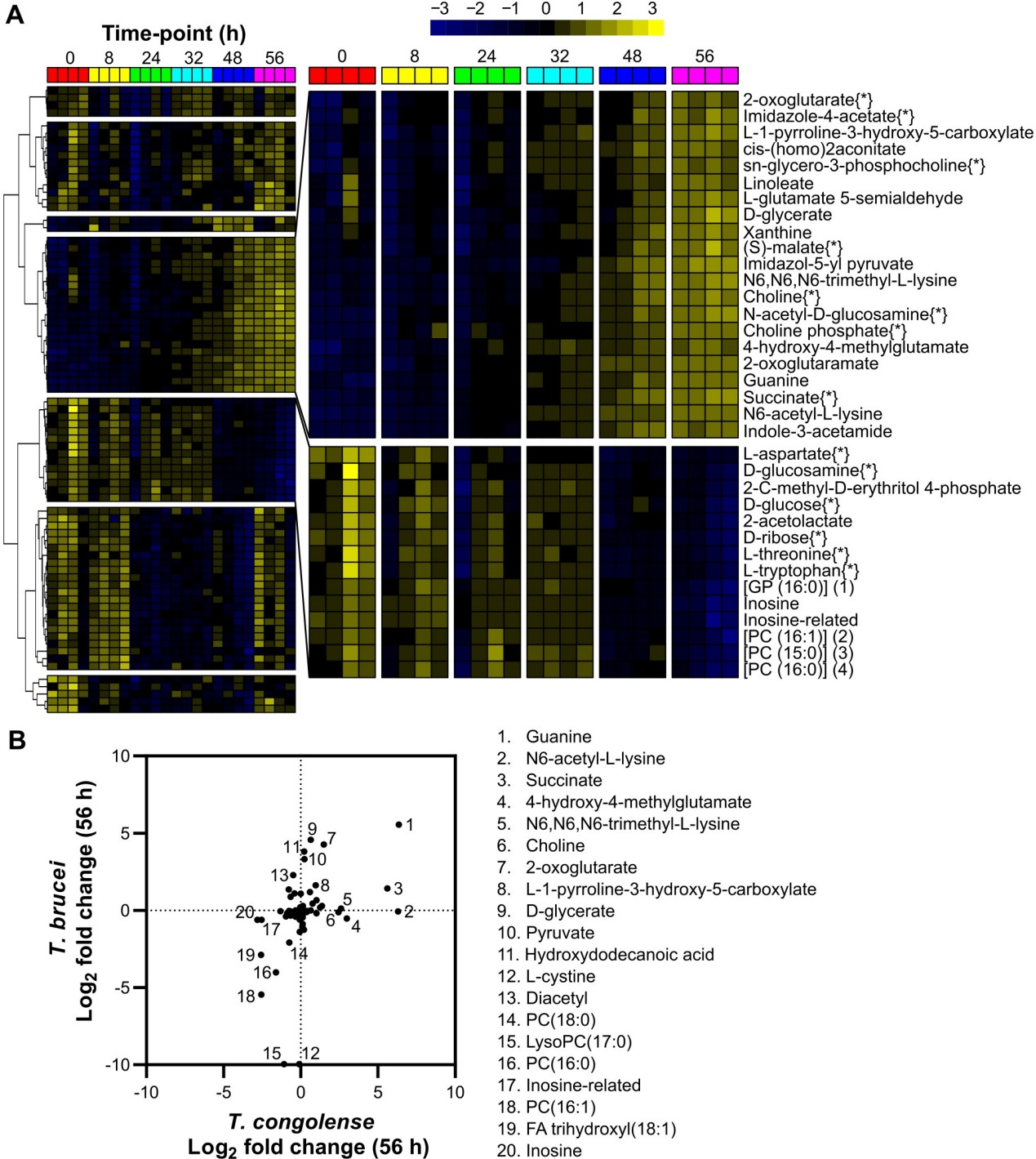

**Fig 2. Analysis of supernatant metabolites after *T. congolense* culture.** A heatmap covering the 80 putative medium components judged to be significantly altered after 56 hours of *in vitro* cell culture containing *T. congolense* strain IL3000, as calculated by a one-way repeated measures ANOVA ($P < 0.05$). Peak abundances were log transformed and mean centred and metabolites were clustered based on Pearson correlation. Two clusters of interest were identified, which are shown in a larger format on the right. Metabolites in the top cluster were observed to increase significantly over time, whilst those in the bottom cluster decreased. Metabolite names followed by {*} were matched to an authentic standard and all other identifications are putative based on mass, retention time and formula. B) Comparison of metabolite changes in medium supernatants after 56 hours between *T. brucei* [60] and *T. congolense* (S3 Table). Relative changes in metabolite abundance were calculated as log$_2$ fold change of 56 h vs 0 h and metabolites exhibiting differences between the species are listed next to the figure.

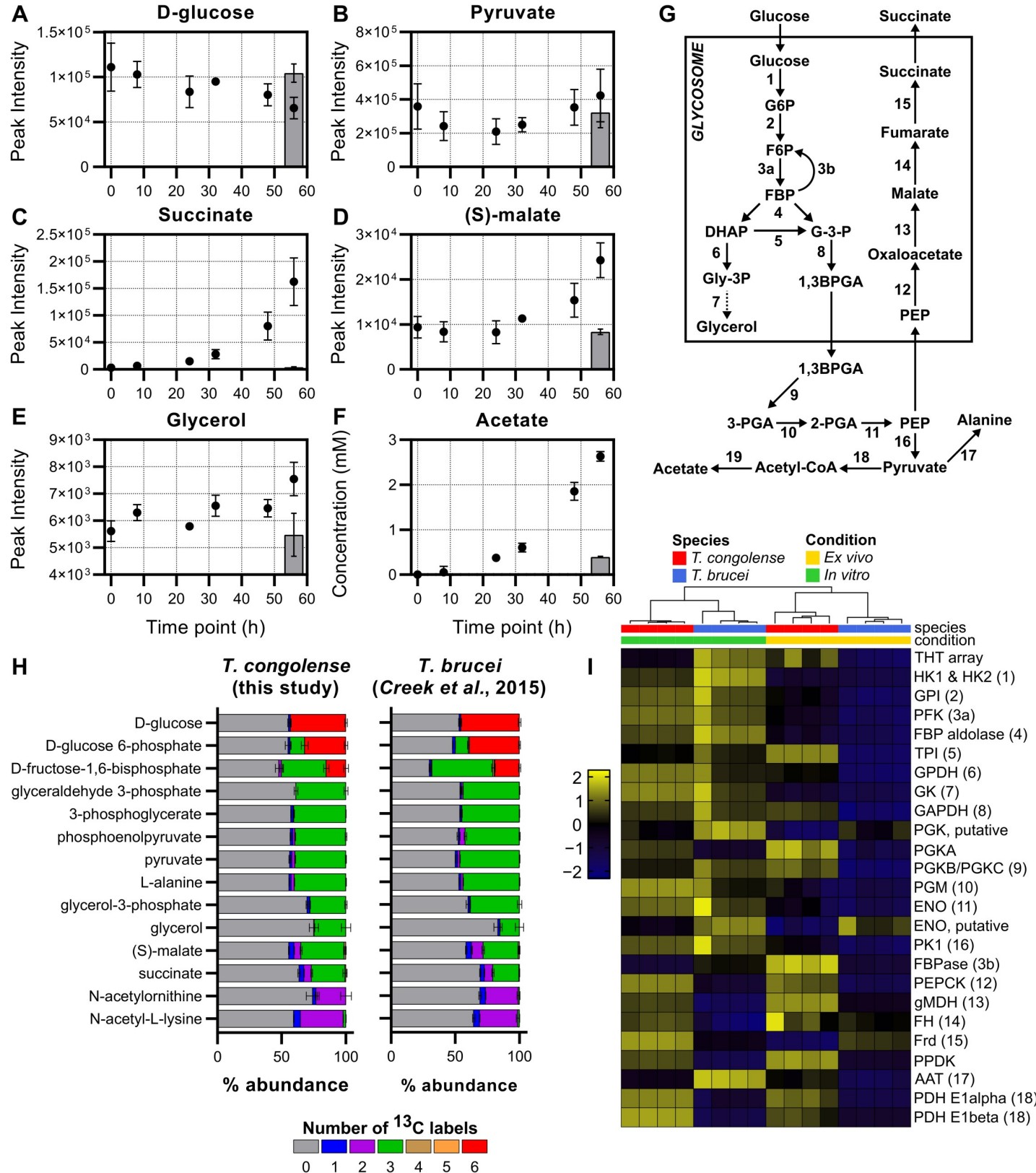

**Fig 3. Energy metabolism in *T. congolense*.** A-E) Supernatant metabolomics analysis of metabolites involved in glycolytic metabolism in *T. congolense*. Grey bars indicate a negative medium control incubated for 56 hours. F) A commercial kit was used to measure acetate concentration during *T. congolense* culture, with

supernatant samples analysed at the same time points as the supernatant metabolomics experiment. G) A simplified overview of the glycolytic pathway. Numbers refer to the following proteins: 1, hexokinase; 2, glucose 6-phosphate isomerase; 3a, phosphofructokinase; 3b, fructose-1,6-bisphosphatase; 4, aldolase; 5, triosephosphate isomerase; 6, glycerol 3-phosphate dehydrogenase; 7, glycerol kinase; 8, glyceraldehyde 3-phosphate dehydrogenase; 9, phosphoglycerate kinase; 10, phosphoglycerate mutase; 11, enolase; 12, phosphenolpyruvate carboxykinase; 13, malate dehydrogenase; 14, fumarate hydratase; 15, NADH-dependent fumarate reductase; 16, pyruvate kinase; 17, alanine aminotransferase; 18, pyruvate dehydrogenase complex; 19, acetate:succinate CoA-transferase and acetyl-CoA thioesterase. H) Tracing glucose derived carbon usage through glycolytic metabolism. *T. congolense* were incubated with a 50:50 mix of $^{12}$C-D-glucose:$^{13}$C-U-D-glucose before cell pellets were isolated for metabolomics analysis. Results were compared to those generated in *T. brucei* by Creek and colleagues [45]. Colours indicate the number of $^{13}$C atoms in each metabolite. I) Comparative analysis of transcript level activity of glycolysis in *T. brucei* and *T. congolense* from both *in vitro* and *ex vivo* conditions. Gene IDs: HK1 & 2, hexokinase, TbTc_0341; GPI, glucose 6-phosphate isomerase, TbTc_1840; PFK, phosphofructokinase, TbTc_1399; ALDA, aldolase, TbTc_0358; TPI, Triosephosphate isomerase, TbTc_1075; GPDH, glycerol 3-phosphate dehydrogenase, TbTc_2722; GK, glycerol kinase, TbTc_0392; GAPDH, glyceraldehyde 3-phosphate dehydrogenase, TbTc_0377; PGK, phosphoglycerate kinase, TbTc_6030; PGKA, phosphoglycerate kinase A, TbTc_0241; PGKB/C, phosphoglycerate kinase B & C, TbTc_0240; PGM, phosphoglycerate mutase, TbTc_5039; ENO1, enolase, TbTc_0465; ENO, putative, enolase, putative, TbTc_3614; PK1, pyruvate kinase 1, TbTc_0372; FBPase, fructose-1,6-bisphosphatase, TbTc_1967; PEPCK, phosphoenolpyrvuate carboxykinase, TbTc_0348; gMDH, glycosomal malate dehydrogenase, TbTc_0642; FH, fumarate hydratase, TbTc_0242; Frd, NADH-dependent fumarate reductase, TbTc_0141; PPDK, pyruvate phosphate dikinase, TbTc_1304; AAT, alanine aminotransferase, TbTc_0675; PDH E1α, pyruvate dehydrogenase E1 alpha subunit, TbTc_4169; PDH E1β, pyruvate dehydrogenase E1 beta subunit, TbTc_5437.

identified based solely on metabolic input and output in *in vitro* culture. An integrated analysis of the metabolomic and transcriptomic datasets was then undertaken in order to further define the metabolic differences between the two species.

## Energy metabolism

As described above, RNA sequencing and culture supernatant metabolomics provided initial indications that BSF *T. congolense* energy metabolism, specifically with respect to glucose usage, diverges substantially from that of BSF *T. brucei* (simplified map of glycolysis depicted in Fig 3G). To dissect metabolic differences at the transcriptome level, pathway analysis was carried out using the TrypanoCyc database [64], which contains 186 manually curated pathways covering 288 genes or groups of multi-copy genes (S4 Table). These analyses showed broadly similar levels of gene expression of glycolytic components between BSF *T. brucei* and *T. congolense* (Fig 3G and 3I). However, the *T. brucei ex vivo* samples displayed a more distinct expression profile, with low transcript abundances for most glycolytic components compared to all sample groups. This is most likely the result of cells being sampled near peak parasitaemia, and as the pleomorphic strain STIB 247 was used, having a higher proportion of tsetse-transmissible, quiescent short stumpy forms–consistent with this there was elevated expression of stumpy markers such as the PAD array (TbTc_0074), PIP39 (TbTc_0700) and reduced expression of RBP10 (TbTc_0619) (S1 Table) [65–67].

Transcripts associated with gluconeogenesis, the succinate shunt, and the acetate generation pathway were increased in abundance in BSF *T. congolense* under both *in vitro* and *ex vivo* conditions compared to BSF *T. brucei*. Key examples of this are PPDK, PEPCK, glycosomal malate dehydrogenase (gMDH) and two subunits of pyruvate dehydrogenase (PDH) (Fig 3I). PPDK was previously reported to be expressed in BSF *T. congolense*, but not in BSF *T. brucei* [47], and it may be assumed that the enzyme serves a similar function in BSF *T. congolense* as it does in PCF *T. brucei*; primarily in a glycolytic role to maintain ATP/ADP balance in the glycosome. The high levels of gMDH expression in BSF *T. congolense* contrasts with BSF *T. brucei*, where gMDH expression is reported to be mostly absent, and cytosolic MDH (cMDH) is the major isoform [68]. The RNAseq analysis also supports a previous study showing high levels of glycerol kinase expression in BSF *T. congolense* [48].

To test whether elevated levels of succinate and malate seen in *T. congolense* supernatant medium samples originated from glucose, LC-MS analysis using $^{13}$C-U-D-glucose was carried out on intracellular metabolites isolated from *in vitro*-cultured cells. Stable isotope analysis has provided valuable insights into *T. brucei* central carbon metabolism [45], and generating *T. congolense* datasets enabled comparative analysis of glucose catabolism (albeit with an

unavoidable difference in medium supplementation of goat serum for *T. congolense*, rather than foetal bovine serum for *T. brucei*).

BSF *T. congolense* was grown for 48 hours in a custom medium (SCM-6; S5 Table), containing a total D-glucose concentration of 10 mM in a 1:1 ratio of D-glucose:$^{13}$C-U-D-glucose prior to LC-MS analysis. Labelling ratios of downstream metabolites were largely similar to that of intracellular glucose, and carbon labelling patterns matched those that would be expected in the BSF *T. brucei* glycolytic pathway (i.e. three $^{13}$C atoms in all metabolites downstream of glyceraldehyde 3-phosphate and glycerol-3-phosphate). Similar to *T. brucei*, a high percentage of 3-carbon labelled fructose-1,6-bisphosphate (FBP) (34.8%) was observed in *T. congolense* (Fig 3H), probably a result of the "reverse" aldolase reaction occurring in the glycosome [45]. Importantly, two-carbon labelling was observed in several acetylated compounds (N-acetylornithine & N-acetyl-L-lysine; Fig 3H), confirming that acetyl groups used to generate these metabolites originate from D-glucose. Although acetyl-CoA, the product of pyruvate oxidation, was not detected for technical reasons, the 2-carbon labelling patterns of acetylated metabolites suggests that the acetyl moieties in these compounds originate from glucose, potentially through acetyl-CoA and acetate, as previously evidenced in other trypanosomatids [45,69]. Taken together, these data indicate that the flow of carbon atoms for glycolytic components in *T. congolense* is very similar to that in *T. brucei*. However, the metabolic outputs differ drastically from BSF *T. brucei* and appear to be more similar to PCF *T. brucei*.

Metabolite labelling was corrected for the 1:1 (50%) ratio of natural glucose to $^{13}$C-U-D-glucose, which equated to a mean percentage labelling of 43.2% (the value is less than 50% due to D-glucose in the serum). All glycolytic metabolites up to pyruvate showed >40% labelling when corrected (for glucose 6-phosphate and fructose-1,6-bisphosphate, both 3-carbon and 6-carbon labels were taken into account), although glycerol and glycerol 3-phosphate exhibited 28.5% and 32.5% labelling, respectively, as these metabolites can also be obtained from catabolism of lipid precursors. Moreover, 42.4% (49.1% corrected) labelling was detected in L-alanine, suggesting that the alanine aminotransferase reaction that utilizes pyruvate to generate 2-oxoglutarate and L-alanine in both BSF and PCF *T. brucei*, also occurs in BSF *T. congolense* [45,70]. For both malate and succinate, 3 carbons were derived from glucose and these metabolites showed 40.2% (46.6% corrected) and 32.3% (37.7% corrected) labelling, respectively. These results suggest that glucose is not the only source of intracellular succinate and malate in *T. congolense*. However, these values were higher than those reported in *T. brucei* (35% and 26% for malate and succinate, respectively [45]).

Whilst PCF *T. brucei* exhibit TCA cycle activity, this pathway is not used to catabolize glucose [38,40]. No citric acid cycle intermediate isotopologues (e.g. citrate) were found when BSF *T. congolense* were incubated with $^{13}$C-U-D-glucose, indicating that like in *T. brucei*, *T. congolense* TCA metabolism is not linked to glycolysis. However, small amounts of 2-carbon labelled succinate and malate were observed (Fig 3H), which can be explained by the reversal of the glycosomal succinate shunt. Taken together, these data suggest that BSF *T. congolense*, both from *in vitro* cultures and *in vivo* infection, metabolically resemble an intermediate between BSF and PCF *T. brucei*, with moderate glycolytic capacity and significant levels of succinate shunt activity (glycosomal, rather than mitochondrial; S1 Table) as well as a highly active mitochondrial acetate generating pathway.

Previous work has shown that reduction of glucose concentrations in BSF *T. brucei* culture from 10 mM to 5 mM leads to decreased cellular motility, reduction in growth and cell body rounding morphology within 8 hours [71]. Given that glucose was not substantially depleted in *T. congolense* cultures after 56 h, we tested the effect of reduced glucose concentrations on BSF *T. congolense* viability. Unlike *T. brucei*, *T. congolense* was able to maintain a growth rate equal to controls at concentrations as low as 2 mM (Fig 4A) when continuously passaged with

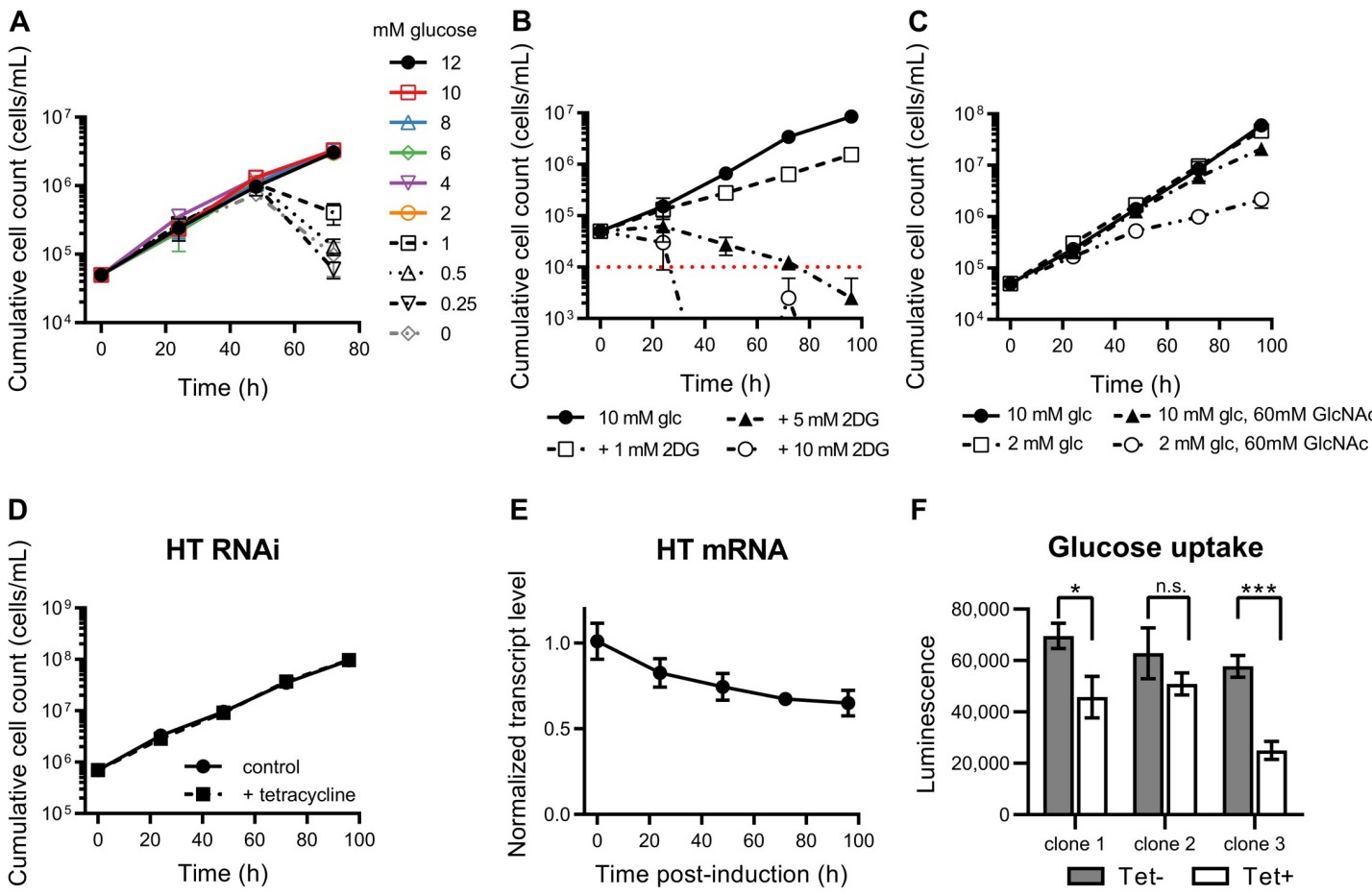

**Fig 4. *In vitro* analysis of glycolytic metabolism.** A) *T. congolense* remains viable in reduced glucose concentrations. A growth defect was only observed when glucose concentrations were reduced to <2 mM. B) Supplementation with increased concentrations of 2-deoxy-D-glucose leads to *T. congolense* cell death (red dotted line indicates detection limit by haemocytometer). C) Analysis of growth in the presence and absence of N-acetyl-D-glucosamine. Parasites were cultured in SCM-6 supplemented with 10 mM or 2 mM glucose in the presence or absence of 60 mM GlcNAc and density monitored by haemocytometer every 24 hours. D) Knock-down of the entire glucose transporter (TcoHT) array does not affect *in vitro* cell viability. RNAi was induced in three independent clones by the addition of 1 μg/mL tetracycline (Tet), and cell densities of induced and uninduced cells were monitored daily. E) Normalised TcoHT mRNA abundance over time after RNAi induction. F) Changes in glucose uptake in RNAi-induced cells were detected via an enzyme-linked luminescence assay coupled to 2-deoxy-D-glucose uptake over a period of 30 minutes. The assay was carried out 72-hours post-induction. Of the three RNAi lines, 2 showed a significant reduction in glucose uptake capability (Student's T-test, $^{*}P < 0.05$; $^{***}P < 0.001$).

no observable change in morphology or motility. To test whether glucose-derived ATP was essential to *T. congolense*, cells were incubated with D-glucose in addition to varying concentrations of 2-deoxy-D-glucose (2DG), which can be internalised, but not metabolised further than 2-deoxy-D-glucose 6-phosphate. As a result, glycolysis and pentose phosphate pathway metabolism are inhibited and glycosomal ATP levels are depleted as they cannot be regenerated in the latter stages of glycolysis, leading to a drastic change in glycosomal ATP/ADP ratio [72] (Fig 4B). Incubation of *T. congolense* in medium supplemented with 2DG (in addition to 10 mM glucose) led to growth defects in a dose dependent manner, likely due to 2DG being outcompeted by glucose at lower concentrations (Fig 4B). Although the growth defect was minor in the presence of 1 mM 2DG, there was a more pronounced reduction with 5 mM 2DG. When equimolar concentrations of glucose and 2DG were used, growth was repressed and cell death occurred within 48 hours (Fig 4B). *T. congolense* viability was also tested in SCM-6 in the presence of N-acetyl-D-glucosamine (GlcNAc), a sugar that inhibits glucose

uptake [73] (Fig 4C). In the presence of 60 mM GlcNAc with 10 mM glucose, there was a moderate, yet significant (p < 0.0001 at 96 h, *t*-test of cell densities) growth defect in *T. congolense* (Fig 4C). Viability was further reduced when the same concentration of GlcNAc was used alongside 2 mM glucose (p < 0.0001 at 96 h, *t*-test of cell densities), the lowest concentration *T. congolense* could tolerate (Fig 4C). The rate of glucose consumption was measured by assaying glucose concentrations in cell culture supplemented with 4 mM glucose. Doubling times (6–7 hours and 11–12 hours for *T. brucei* and *T. congolense*, respectively) were taken into account, and rate of glucose uptake was shown to be $47.17 \pm 27.91$ nmol$^{-1}$ min$^{-1}$ $10^8$ cells in *T. congolense*, significantly lower than the rate ($132.18 \pm 16.31$ nmol$^{-1}$ min$^{-1}$ $10^8$ cells) in *T. brucei* (n = 4, p = 0.0039; *t*-test) which was comparable to previous studies [27,61].

To further probe the essentiality of glucose uptake in BSF *T. congolense*, RNAi was used to knock down expression of the hexose transporter (TcoHT) array, specifically those matching the THT1 and THT2 array in *T. brucei* (TcIL3000.A.H_000260500, TcIL3000.A.H_000260600, TcIL3000.A.H_000794500, TcIL3000.A.H_000794600, TcIL3000.A.H_000794700), which has been shown to significantly restrict growth of BSF *T. brucei* [74]. Whilst growth rate was unaffected in BSF *T. congolense* (Fig 4D), induction of HT RNAi led to a reduction in transcript abundance at all time points (mean transcript levels of 83%, 75%, 68% and 65% compared to uninduced controls at 24, 48, 72 and 96 h post-induction, respectively; Fig 4E). Glucose uptake was decreased (mean reduction of 37% in uptake compared to uninduced controls after 72 h; Fig 4F), suggesting that either lower levels of glucose are sufficient for energy generation in *T. congolense*, or the parasite can utilize other carbon sources for ATP production. These alternative sources could include medium or serum components such as amino acids [75].

PCF *T. brucei* express most components of the electron transport chain (ETC) to generate ATP through oxidative phosphorylation. In contrast, BSF *T. brucei* express an $F_1F_o$-ATPase that functions in reverse, and alternative oxidase [76]. In addition, a recent study has suggested that complex I is expressed and functional in BSF *T. brucei* [77]. Transcriptomics analysis of the ETC was carried out, using a gene list generated by Zikova and colleagues [76], but no significant patterns could be discerned, and thus we were not able to draw a conclusion with regards to ETC activity in BSF *T. congolense* based on transcriptomics data alone (S1 Table and S3 Fig).

## Nucleotide metabolism

Metabolomic analysis of BSF *T. congolense* culture supernatants indicated a significant uptake of exogenous ribose, a contributor to nucleotide metabolism via uptake, or via the pentose phosphate pathway (PPP; Figs 2A and 5A). Whilst guanosine was not detected in the supernatant, significant accumulation of guanine (Fig 5B) was observed, suggesting either excretion of this metabolite, or, hydrolysis of guanosine through parasite-secreted hydrolases/nucleosidases (previously identified in BSF *T. brucei* secretomes [78,79]). This mechanism would enable uptake of guanine and other nucleobases through nucleobase transporters, for which multiple orthologues have been identified in the *T. congolense* genome [21] through homology with known *T. brucei* nucleobase transporters TbNT8.1 and TbNBT1 [80,81]. In addition, there was an accumulation of xanthine, a product of xanthosine hydrolysis, and depletion of inosine, an important nucleoside composed of hypoxanthine and ribose (Fig 5C and 5D). The nucleoside cytidine and the nucleobase hypoxanthine were also detected, but appeared to remain unchanged during the time course, although the latter was a medium supplement potentially added in excess (S3 Table). It is noteworthy that only a single nucleoside transporter gene (TbTc_1072; *T. congolense* gene IDs: TcIL3000.A.H_000665800 and the pseudogene TcIL3000.A.H_000679300; S2 Table) can be identified in *T. congolense*, which is a syntenic

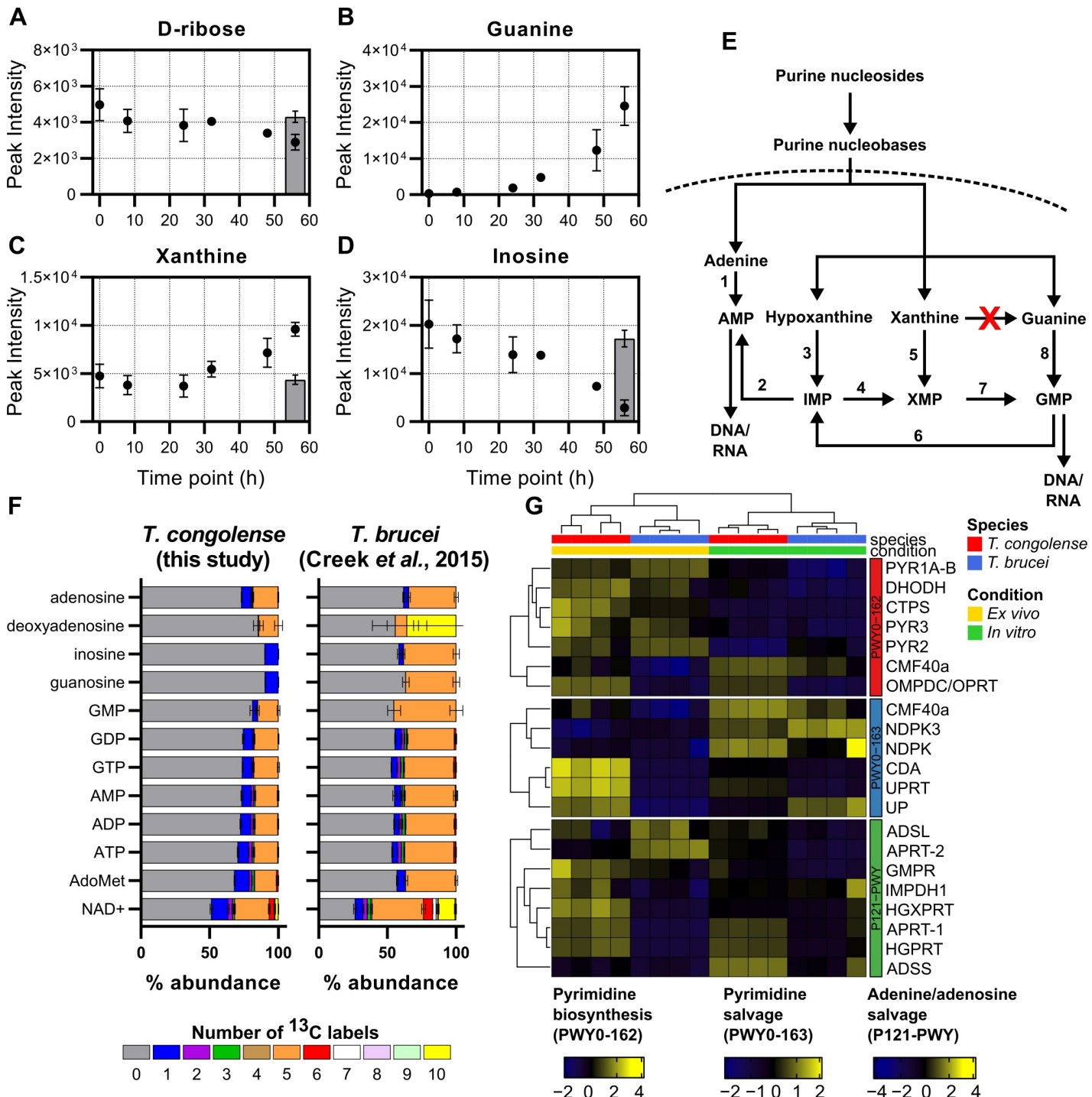

**Fig 5. Nucleotide metabolism in *T. congolense*.** Supernatant analysis of *T. congolense in vitro* cultures showing changes in abundance of D-ribose (A), guanine (B), xanthine (C) and inosine (D) over 56 hours. Grey bar indicates a negative medium control group E) Simplified overview of purine salvage and synthesis in trypanosomatids (adapted from [148]). Numbers indicate the following enzymes: 1, APRT; 2, AD; 3, HGPRT; 4, IMPD; 5, HGXPRT; 6, GMPR; 7, GMPS; 8, HGPRT. Red cross indicates guanine deaminase, which is not encoded/annotated in the *T. congolense* genome (based on current assembly). F) Comparison of glucose-derived purine carbon labelling in *T. congolense* and *T. brucei* [45]. Colours indicate the number of $^{13}$C atoms in each metabolite. G) Comparative RNAseq analysis of *T. congolense* and *T. brucei* under both *in vitro* and *ex vivo* conditions. Gene IDs from top to bottom: PWY0-162 (pyrimidine biosynthesis): PYR1A-B, glutamine hydrolysing carbomoyl phosphate synthase, TbTc_1631; DHODH, dihydroorotate dehydrogenase (fumarate), TbTc_0620; CTPS, cytidine triphosphate synthase, TbTc_0920; PYR3, dihydroorotase, TbTc_3801; PYR2, aspartate carbamoyltransferase, TbTc_1630; CMF40a, nucleoside diphosphate kinase, TbTc_5784; OMPDC/OPRT, orotidine-5-monophosphate decarboxylase/orotate phosphoribosyltransferase, TbTc_0735. PWY0-163 (pyrimidine salvage): CMF40a, nucleoside diphosphate

kinase, TbTc_5784; NDPK3, nucleoside diphosphate kinase 3, TbTc_2560; NDPK, nucleoside diphosphate kinase, TbTc_0593; CDA, cytidine deaminase, TbTc_3318; UPRT, uracil phosphoribosyltransferase, TbTc_4220; UP, uridine phosphorylase, TbTc_5794. P121-PWY (adenine/adenosine salvage): ADSL, adenylosuccinate lyase, TbTc_1986; APRT-2, glycosomal adenine phosphoribosyltransferase, TbTc_5918; GMPR, GMP reductase, TbTc_4627; IMPDH1, inosine-5'-monophosphate dehydrogenase, TbTc_1648; HGXPRT, hypoxanthine-guanine-xanthine phosphoribosyltransferase, TbTc_3696; APRT-1, cytosolic adenine phosphoribosyltransferase, TbTc_3522; HGPRT, hypoxanthine-guanine phosphoribosyltransferase, TbTc_0726; ADSS, adenylosuccinate synthetase, TbTc_1142.

homologue of TbNT10 [21], functionally characterized as a P1-type purine nucleoside transporter [82], and is thus unlikely to transport cytidine [83].

Purine salvage is an essential process in trypanosomatids, as they lack the *de novo* synthesis pathway for the purine ring [84], and previous analysis of cell pellets to investigate intracellular nucleotide metabolism utilizing $^{13}$C-U-D-glucose in BSF *T. brucei* showed purine salvage pathways incorporating 5-carbon labelled ribose derived from glucose [45] (Fig 5F). Whilst the ribose incorporated into these nucleosides originates almost exclusively from glucose in *T. brucei* (Fig 5F), *T. congolense* appears to use far less glucose-derived ribose to make purine nucleosides such as adenosine, guanosine and inosine (Fig 5F).

Transcriptomics analyses indicated upregulation of genes associated with generation of adenosine nucleotides (Fig 5G; red vertical bar), especially in *ex vivo T. congolense*, as well as hypoxanthine-guanine phosphoribosyltransferase (HGPRT) and uracil phosphoribosyltransferase (UPRT). Upregulation of nucleoside hydrolases and phosphoribosyltransferases supports a previous suggestion based on genome content that *T. congolense* has a capacity for nucleobase uptake [21].

The purines guanosine and inosine, which incorporate glucose-derived ribose in *T. brucei*, were almost entirely unlabelled in *T. congolense* (Fig 5F). However, the phosphorylated nucleosides GMP, GDP and GTP all incorporate glucose-derived carbon atoms, presumably through ribose. Given the labelling patterns seen in adenosine, one possible explanation could be conversion of AMP to inosine monophosphate (IMP; adenosine monophosphate deaminase; TbTc_0145), IMP to xanthosine monophosphate (IMP dehydrogenase; TbTc_1648) and XMP to GMP (GMP synthase; TbTc_1452). However, only one of these enzymes, GMP synthase, was expressed at higher abundance in *T. congolense* (log$_2$ fold change: 1.61 and 2.06 for *ex vivo* and *in vitro*, respectively; false discovery rate-adjusted $p < 0.001$). Overall, incorporation of glucose-derived carbons into purine nucleosides is reduced in *T. congolense* compared to *T. brucei*. It should be noted that in both experiments, there was no ribose supplementation in the media

Of the pyrimidines, uracil and its derivatives were detected during the glucose labelling experiment (S4 Fig). Uracil is known to be the main pyrimidine salvaged by other kinetoplastids including *T. brucei* [85–87]. Whilst the majority of the uridine, UMP, UDP and UTP pools incorporate glucose-derived ribose (five $^{13}$C labels), 5-carbon isotopologues of these pyrimidines were reduced in abundance in *T. congolense* compared to *T. brucei*. Conversely, 2-carbon labelled isotopologues appeared to comprise the majority of uridine, uracil and their nucleotides (S4 Fig).

Whilst uracil biosynthesis is not essential in *T. brucei* [88], the uracil pool in *T. congolense* appears to derive almost entirely from glucose, when corrected for 50% glucose labelling (45.9% in *T. congolense* vs 24% in *T. brucei* [45]; S4 Fig), suggesting that this species predominantly synthesizes uracil from orotate to UMP (orotate phosphoribosyltransferase/orotidine 5-phosphate decarboxylase; TbTc_0735) and from UMP to uracil (uracil phosphoribosyltransferase; TbTc_4220), as can occur in *T. brucei* [45]. Both these genes are expressed at higher abundance in *T. congolense*, both *in vitro* and *ex vivo*, compared to *T. brucei* (Fig 5G and S1 Table), which could explain the increased isotopologue labelling.

These data indicate that, at least under the growth conditions used here, BSF *T. congolense* favours purine nucleoside/nucleotide synthesis from nucleobases with a reduced dependence on glucose-derived ribose 5-phosphate, in addition to *de novo* synthesis of orotate, uracil and uridine nucleosides. However, the difference in serum requirements for the two organisms is a confounding factor to the interpretation of this difference.

## Amino acid metabolism

It is well established that trypanosomatid parasites scavenge amino acids, key nutrients for survival, from their hosts [89,90]. Therefore, comparative analyses of *T. congolense* and *T. brucei* amino acid metabolism were undertaken. Whilst the majority of amino acids were detected during the supernatant time course, relative abundances in the medium did not vary greatly after 56 hours of *in vitro* culture (Fig 6A–6C and S3 Table). The greatest reductions were observed in threonine ($\log_2$ FC after 56 hours: -0.89; Fig 6A), tryptophan ($\log_2$ FC: -0.74; Fig 6B), glutamine ($\log_2$ FC: -0.39), asparagine ($\log_2$ FC: -0.35) and phenylalanine ($\log_2$: -0.35). Interestingly, cysteine, an essential factor for the *in vitro* culture of *T. brucei* [63,91], was not significantly consumed by 56 hours ($\log_2$ FC: -0.07; Fig 6C). However, at least low-level exogenous cysteine is still required to sustain parasite growth *in vitro*, as viability was significantly affected in the absence of cysteine (for both 1.5 mM and 1 mM vs 0 mM cysteine, $p < 0.0001$, *t*-test of cells densities at 96 h; S5 Fig). Experiments were carried out to test the essentiality of all other individual amino acids (with the exception of glutamine, known to be an important amino donor in trypanosomatid metabolism [60,89]). Using the minimal medium SCM-6, cell viability was monitored for 72 hours in the absence of specific amino acids. Removal of the following amino acids from culture medium led to defects in growth over 72 hours: asparagine, histidine, isoleucine, leucine, methionine, proline, serine, tyrosine and valine (Fig 6D–6G). Whilst aspartate appeared to be depleted in spent culture supernatants (S3 Table), this also occurred in the medium only control. Furthermore, removal of aspartate did not lead to reduced cell viability or growth rate in culture (Fig 6F). Long term culture was impossible without the addition of phenylalanine and threonine, leading to a final culture formulation, SCM-7 (S5 Table), containing a total of 14 amino acids. Therefore, BSF *T. congolense* appears to require a higher number of amino acids than BSF *T. brucei*, at least *in vitro*, with CMM containing only 8 amino acids in total, including cysteine and glutamine [60]. To further probe amino acid metabolism, pathway analysis was carried out on the transcriptome (S6 Fig) and metabolome (Figs 6 and S7 and S8).

BSF *T. brucei* utilizes exogenous L-glutamine as the primary source of intracellular glutamate and 2-oxoglutarate, and produces significant levels of glutamine-derived succinate [45,89] (Fig 6I). Given the high levels of succinate excreted by *T. congolense*, stable isotope labelling was used to determine the contribution of L-glutamine to this pool. *T. congolense* was incubated for 48 hours with 1 mM $^{13}$C-U-L-glutamine and cell pellets were analysed by LC-MS. Results indicated the presence of biochemical activities consistent with those previously observed in *T. brucei* [89] (Fig 6). Significant glutamine-derived carbon labelling was detected after 48 h incubation for succinate (41.3%, 48.5% corrected), glutamate (76.1%, 89.2% corrected), 2-oxoglutarate (80.5%, 94.3% corrected) and succinate semialdehyde (94.7% corrected, Fig 6I). As would be anticipated, labelling of glutathione (86.1%) and trypanothione (98.4%) from glutamine through glutamate was also observed (S7 Fig). No labelling of malate or aspartate was seen in this study, despite the use of high concentrations of $^{13}$C-U-L-glutamine compared to the equivalent study performed in *T. brucei* with a 50:50 ratio of $^{13}$C-U-L-glutamine [89]. Transcriptomics analysis showed high expression levels of glutamine synthetase in *T. congolense*, compared to *T. brucei*, under *ex vivo* conditions only, suggesting ATP-

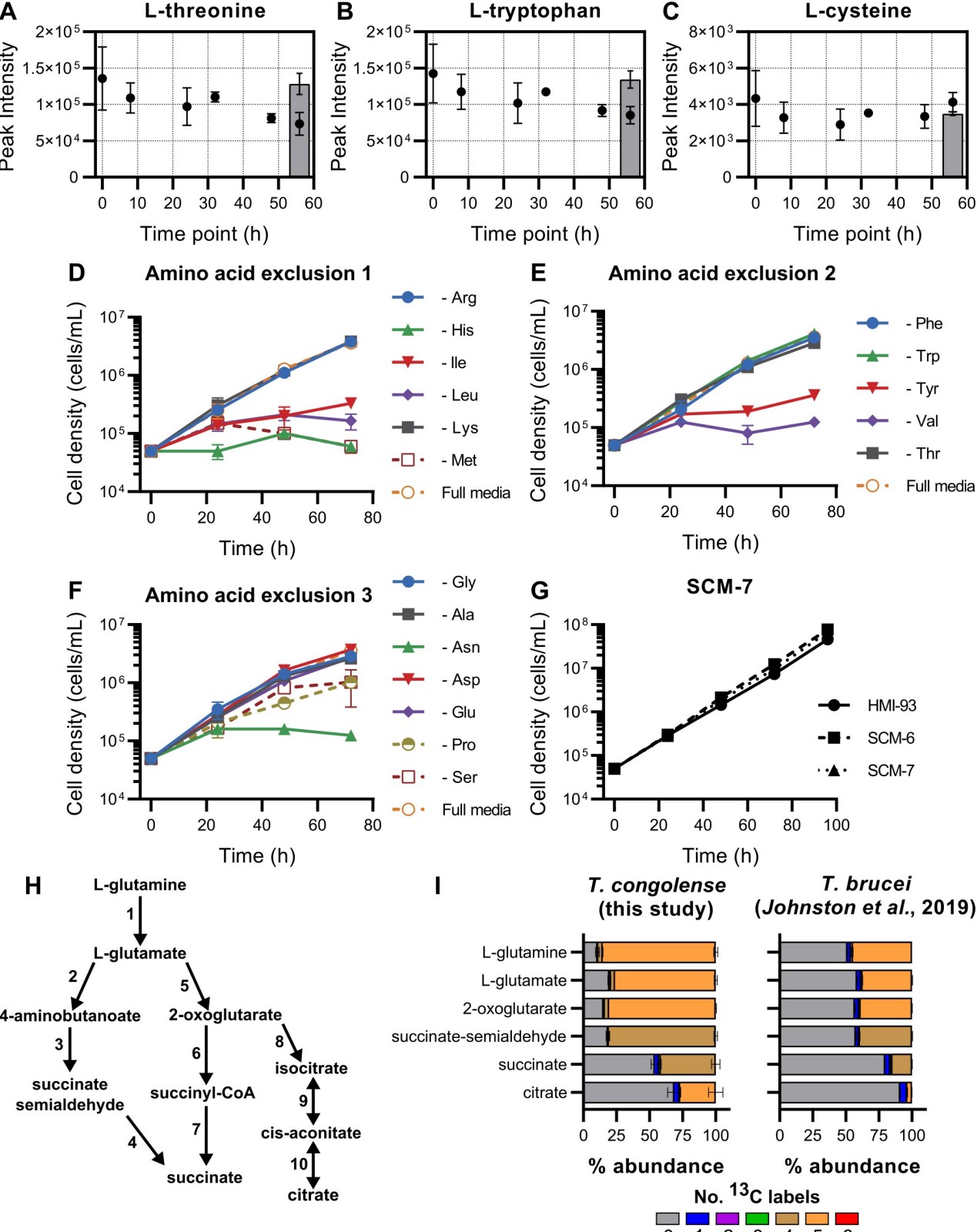

**Fig 6. Amino acid metabolism in *T. congolense* IL3000.** A-C) Analysis of indicated amino acids in *T. congolense* IL3000 culture supernatants over a 56 h time course. Grey bars indicate a negative medium control group. D-F) Growth curves in SCM-6 excluding one amino acid at a time, to

determine those essential to *T. congolense* viability. In each experiment, full SCM-6 was used as a positive control. Legends indicate which amino acid was removed in each experiment. G) Growth analysis of SCM-6 and SCM-7, the latter containing only amino acids deemed essential, compared to HMI-93 [124]. H) Simplified map of intracellular glutamine metabolism. Numbers refer to the following enzymes: 1, glutaminase; 2, glutamate decarboxylase; 3, 4-aminobutyrate aminotransferase; 4, succinate semialdehyde dehydrogenase; 5, glutamate dehydrogenase; 6, 2-oxoglutarate dehydrogenase; 7, Succinyl-CoA synthetase; 8, isocitrate dehydrogenase; 9 & 10, aconitase. I) Carbon utilisation from L-glutamine was analysed in *T. congolense* (100% $^{13}$C-U-L-glutamine) and compared to that in *T. brucei* (50:50 ratio of L-glutamine and $^{13}$C-U-L-glutamine) [89].

dependent generation of glutamine may occur in the parasite under these conditions (S1 Table).

The apparent essentiality of several amino acids was also investigated using stable isotope labelling. Proline is an essential carbon source for PCF *T. brucei* and is required for protein synthesis in BSF *T. brucei*, although it is not a required supplement in BSF medium [92]. In contrast, removal of proline from BSF *T. congolense* medium led to reduced growth (Fig 6F). RNAi-mediated knock-down of proline metabolism (specifically pyrroline-5-carboxylate dehydrogenase, TbP5CDH) in PCF *T. brucei* has highlighted the requirement of proline metabolism for mitochondrial function [92]. Indeed, both P5CDH (TbTc_1695) and proline dehydrogenase (TbTc_1591) were expressed at higher levels in *ex vivo T. congolense*, compared to *T. brucei* (S1 Table and S6 Fig). However, $^{13}$C-U-L-proline labelling showed that this amino acid did not contribute meaningfully to the biosynthesis of other metabolites (S8 Fig). Therefore, similar to BSF *T. brucei*, a requirement for proline in BSF *T. congolense* may be for the purposes of polypeptide synthesis only.

As in *T. brucei* [45], glucose-derived carbon usage was detected in several amino acids in *T. congolense* (S6A Fig). Aspartate (a precursor for pyrimidine nucleotide biosynthesis) and alanine (a by-product of a pyruvate-utilising aminotransferase reaction) (S6A Fig) exhibited 3-carbon isotopologues derived from $^{13}$C-U-D-glucose in both species. However, in *T. brucei*, a small proportion of L-asparagine labelling was observed (1.2% 3-carbon labelling) [45], whilst none was observed in *T. congolense* (S6A Fig). The metabolism of asparagine has not been studied in African trypanosomes; given the reduction of cell growth in the absence of this amino acid (Fig 6F), labelling with $^{13}$C-U-L-asparagine was performed, but no other labelled metabolites were detected (S8 Fig). This indicates that, as with proline, asparagine uptake is required principally for protein synthesis in *T. congolense*. The reduced expression of asparagine synthetase (TbTc_4894; TcIL3000.A.H_000497800), which converts aspartate to asparagine (S6 Fig), suggests that BSF *T. congolense* may rely upon scavenging of exogenous asparagine.

Serine was also shown to be essential to *T. congolense* (Fig 6F), in contrast to minimal culturing requirements for *T. brucei* [60]. $^{13}$C-U-L-serine labelling indicated that *T. congolense* L-serine metabolism mirrors that of *T. brucei* in several aspects, such as *de novo* sphingolipid biosynthesis, with 70.0% 2-carbon labelling of sphinganine and downstream labelling of ceramide and sphingomyelin species (S8 Fig). Similarly, phosphatidylserine decarboxylase activity was evidenced at both transcript and metabolite levels, with 40.1% 2-carbon labelling of glycerol-phospho-ethanolamine (S1 Table and S8 Fig). However, L-serine also has a minor role in S-adenosyl-L-homocysteine detoxification in *T. congolense*, where serine-derived carbon ultimately contributes to cysteine biosynthesis (S8 Fig). Serine-derived carbon labelling was detected in cystathionine (18.1%) and cysteine (16.7%), through to glutathione (4.1%) and trypanothione disulfide (3-carbon labelled, 6.8%; 6-carbon labelled, 0.02%; S7 Fig). Therefore, the inability to exclude L-serine from *T. congolense in vitro* culture media may primarily be attributable to lipid metabolism and an increased demand for serine-derived cysteine, potentially over exogenously obtained cysteine, depending on bioavailability. Indeed, metabolomics

analysis of culture medium indicates that the ability of *T. congolense* to take up cysteine from its environment may be lower than in *T. brucei* (Fig 6C).

Although L-cysteine is primarily considered a source of sulphur for trypanosomatids [89,91,93], we also investigated the carbon contribution of this amino acid in *T. congolense*, and in particular, whether L-cysteine-derived carbon atoms contribute to the biosynthesis of glutathione and trypanothione. $^{13}$C-U-L-cysteine stable isotope labelling experiments were performed (S7 and S8 Figs). Direct replacement of the 1.5 mM L-cysteine present in SCM-6 with $^{13}$C-U-L-cysteine led to high levels of labelling in glutathione and trypanothione disulfide (S7B Fig). This indicates that *T. congolense* can readily take up and metabolise exogenous cysteine, even though abundance of the amino acid is not reduced significantly over 56 hours of parasite *in vitro* culture. Although no clear pattern could be observed in transcriptomic analysis of the trypanothione biosynthesis pathway, both trypanothione synthase (TRYS; TbTc_1359) and trypanothione reductase (TRYR; TbTc_4239) were expressed at high levels in *in vitro* *T. congolense* cells relative to *ex vivo* cells, indicating that under *in vitro* conditions, cells may be subjected to higher levels of oxidative stress (S7C Fig).

There were several further notable observations in the transcriptomics data regarding deamination and decarboxylation of amino acids. For example, cytosolic aspartate aminotransferase (cASAT; TbTc_0799) was expressed at similar levels in both species, indicating that the transamination of tryptophan, tyrosine and phenylalanine to generate aromatic ketoacids [94,95] likely occurs in *T. congolense*. The mitochondrial isoform (mASAT; TbTc_0877) was expressed at higher levels in cultured *T. congolense* (log$_2$ FC: 1.73, p < 0.001), indicative of increased mitochondrial amino acid metabolism.

## Fatty acid metabolism in *T. congolense*

Lipids have a variety of crucial roles in trypanosomes, in particular as a major constituent of membranes. BSF *T. brucei* require large quantities of myristic acid in particular, for the synthesis of glycosylphosphatidylinositol (GPI) that anchors the parasite's major surface glycoproteins [96]. To do this, BSF *T. brucei* both synthesises and scavenges myristic acid. Glucose labelling experiments in *T. brucei* have shown that myristic acid is partially synthesized from glucose-derived carbon through acetyl-CoA, using a system of fatty acid elongases [97] (Fig 7A). One previous study suggested that GPI anchors in *T. congolense* also incorporate myristic acid [98]. However, no saturated fatty acid carbon labelling was detected after incubation of *T. congolense* with $^{13}$C-U-D-glucose (Fig 7A), unlike *T. brucei* [45]. Carbon dissemination was also investigated from threonine, which is used as a source of acetate, and thus, lipids [99] (Fig 7B). Similarly, no saturated lipid carbon labelling was observed, suggesting that *T. congolense* either uses alternative sources of carbon for lipid biosynthesis, or does not rely on acetate as a source of lipids in the same way as *T. brucei* [33].

Several genes associated with acetate/acetyl-CoA metabolism were highly expressed in *T. congolense* compared to *T. brucei* (Fig 7C). For example, ASCT (TbTc_0236), which catalyses ATP-coupled acetate production, was higher under both conditions in *T. congolense* (log$_2$ fold changes: 2.04 and 0.46 for *in vitro* and *ex vivo*, respectively, p < 0.001; S1 Table). Conversely, acetyl-CoA hydrolase (also known as acetyl-CoA thioesterase, ACH; TbTc_5515), an enzyme involved in ATP synthesis-uncoupled acetate production in *T. brucei* [100] was expressed at lower levels in *ex vivo* *T. congolense* compared to *T. brucei* (Fig 7C). Consistent with metabolic data, expression of acetyl-CoA synthetase (AceCS; TbTc_0318), a key enzyme in lipid biosynthesis from acetate, was reduced in both *ex vivo* and *in vitro* *T. congolense* (Fig 7C). Other enzymes involved in fatty acid biosynthesis, namely acetyl-CoA carboxylase (ACC; TbTc_0754), β-ketoacyl-CoA synthase (BKS; TbTc_3372) and β-ketoacyl-CoA reductase

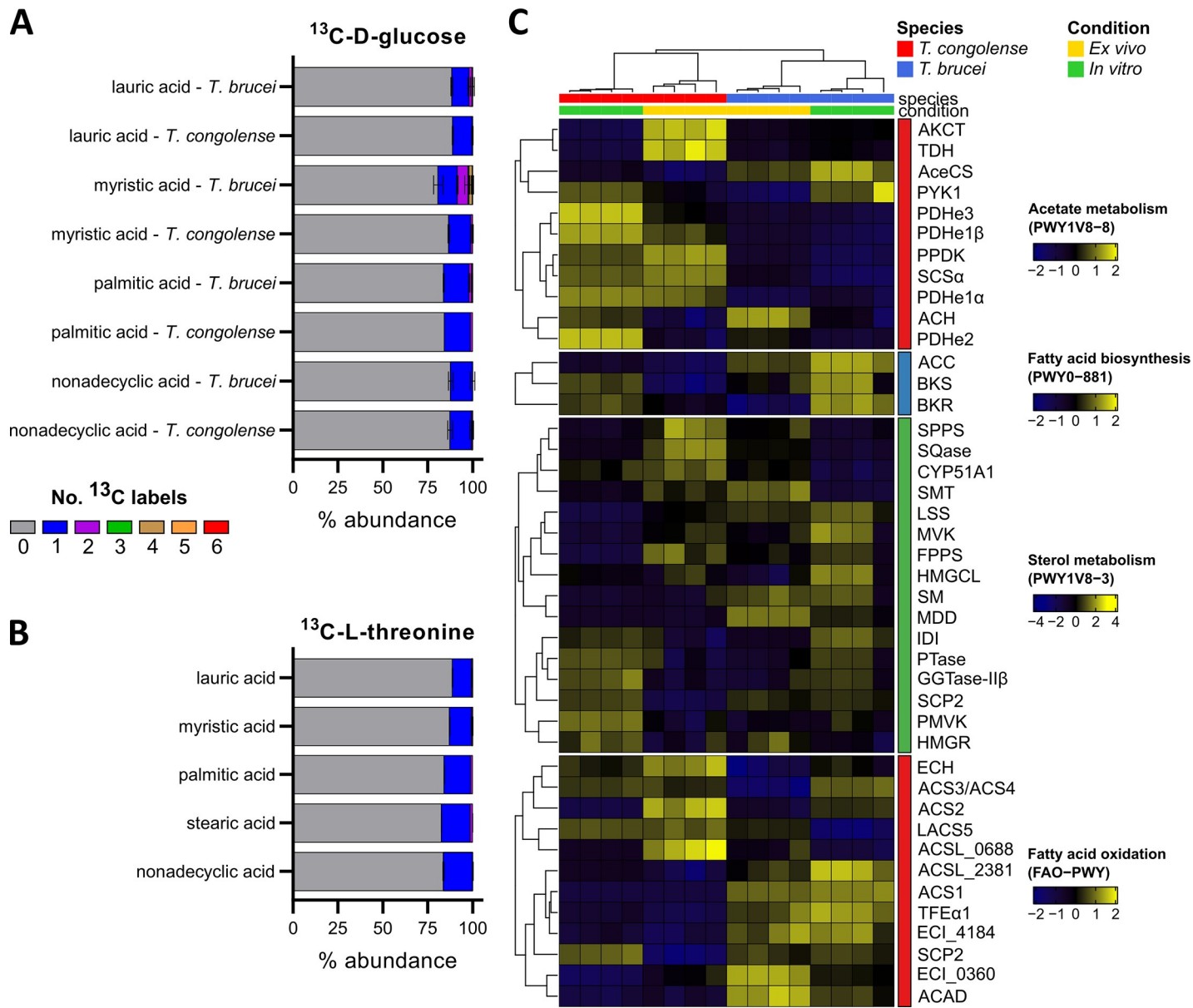

**Fig 7. Fatty acid metabolism in *T. congolense*.** A) Glucose-derived [13]C carbon labelling of saturated fatty acids in *T. congolense* and *T. brucei* [45]. Colours correspond to the number of [13]C labels detected in each metabolite. B) L-threonine-derived saturated fatty acid [13]C labelling in *T. congolense*. Fatty acid systematic names and numbers: lauric acid: dodecanoic acid, C12:0; myristic acid: tetradecanoic acid, C14:0; palmitic acid: hexadecanoic acid, C16:0; nonadecyclic acid: nonadecanoic acid, C19:0. C) Transcriptomics analysis of acetate and lipid metabolism. Gene names and IDs: Acetate metabolism (PWY1V8-8): AKCT, 2-amino-3-ketobutyrate-CoA ligase, TbTc_6236; TDH, L-threonine 3-dehydrogenase, TbTc_5991; AceCS, acetyl-CoA synthetase, TbTc_0318; PYK1, pyruvate kinase, TbTc_0372; PDHe3, pyruvate dehydrogenase E3, TbTc_4765; PDHe1β, pyruvate dehydrogenase E1 β subunit, TbTc_5437; PPDK, pyruvate phosphate dikinase, TbTc_1304; SCSα, succinyl-CoA synthetase α subunit, TbTc_0813; PDHe1α, pyruvate dehydrogenase E1 α subunit, TbTc_4169; ACH, acetyl-CoA hydrolase/thioesterase, TbTc_5515; PDHe2, dihydrolipoamide acetyltransferase, TbTc_1015. Fatty acid biosynthesis (PWY0-881): ACC, acetyl-CoA carboxylase, TbTc_0754; BKS, β-ketoacyl synthase, TbTc_3372; BKR, β-ketoacyl-ACP reductase, TbTc_1241. Sterol metabolism (PWY1V8-3): SPPS, solanesyl-diphosphate synthase, TbTc_3025; SQase, squalene synthase, TbTc_2577; CYP51A1, lanosterol 14α demethylase, TbTc_4837; SMT, sterol 24-c methyltransferase, TbTc_0387; LSS, lanosteral synthase, TbTc_4540; MVK, mevalonate kinase, TbTc_3761; FPPS, farnesyl pyrophosphate synthase, TbTc_5375; HMGCL, hydroxymethylglutaryl-CoA lyase, TbTc_6160; SM, squalene monooxygenase, TbTc_3357; MDD, mevalonate diphosphate decarboxylase, TbTc_0546; IDI, isopentenyl-diphosphate delta-isomerase, TbTc_1099; PTase, prenyltransferase, TbTc_1352; GGTase-IIβ, geranylgeranyl transferase type II β subunit, TbTc_0680; SCP2, 3-ketoacyl-CoA thiolase, TbTc_4024; PMVK, phosphomevalonate kinase, TbTc_3039; HMGR, 3-hydroxy-3-methylglutaryl-CoA reductase, TbTc_3189. Fatty acid oxidation (FAO-PWY): ECH, enoyl-CoA hydratase, TbTc_3283; ACS3/ACS4, fatty acyl-CoA synthetase 3 & 4, TbTc_0101; ACS2, fatty acyl-CoA synthetase 2, TbTc_0102; LACS5, fatty acyl-CoA synthetase, TbTc_0099; ACSL_0688, long-chain-fatty-acid-CoA ligase, TbTc_0688; ACSL_2381, long-chain-fatty-acid-CoA ligase, TbTc_2381; ACS1, fatty acyl-CoA synthetase 1, TbTc_0100; TFEα1, enoyl-CoA hydratase/enoyl-CoA isomerase, TbTc_3362; ECI_4184, 3,2-trans-enoyl-CoA isomerase, TbTc_4184; SCP2, 3-ketoacyl-CoA thiolase, TbTc_4024; ECI_0360, 3,2-trans-enoyl-CoA isomerase, TbTc_0360; ACAD, acyl-CoA dehydrogenase, TbTc_4954.

(BKR; TbTc_1241), were all expressed at lower abundance in *T. congolense* than *T. brucei*, in particular in *ex vivo* cells (Fig 7C). Of the four elongases, ELO1 (TbTc_0159) and ELO2 (TbTc_1882) were expressed at similar levels in both BSF *T. congolense* and *T. brucei* (S1 Table). Whilst expression of ELO3 (TbTc_0235) appeared to be reduced in *T. congolense* ($\log_2$ FC: -1.53 and -1.93 compared to *T. brucei* for *in vitro* and *ex vivo*, respectively, $p < 0.001$; S1 Table), *T. congolense* cells expressed higher levels of ELO4 (TbTc_0737) in both *in vitro* and *ex vivo* conditions, compared to *T. brucei* ($\log_2$ FC: 1.43 and 1.44 for *in vitro* and *ex vivo* comparisons, respectively, $p < 0.001$).

Lipoic acid, a fatty acid *de novo* synthesised by kinetoplastids [97,101], was not detected using the LC-MS platform. However, gene expression data indicated that transcript abundance of genes involved in lipoic acid synthesis (lipoic acid synthase, TbTc_4472; dihydrolipodamide dehydrogenase, TbTc_0275 and TbTc_0276) was similar across both species ($p = 0.115$), suggesting *T. congolense* also synthesise lipoic acid *de novo*.

The variation in observed gene expression associated with the sterol pathway appeared to correlate with sample condition rather than species (Fig 7C). However, *T. congolense* transcripts for genes involved in lanosterol synthesis were reduced, especially under *in vitro* conditions (squalene synthase, SQase, TbTc_2577; squalene monooxygenase, SM, TbTc_3357; lanosterol synthase, LSS, TbTc_4540; Fig 7C).

Transcripts associated with fatty acid oxidation were less abundant in *T. congolense* compared to *T. brucei* under both conditions (Fig 7C). Energy generation from fatty acids has not been reported for *T. brucei*. In *T. congolense*, high abundance of transcripts in key genes such as enoyl-CoA hydratase (ECH; TbTc_3283), particularly in *ex vivo* samples, was observed. It would therefore be important to establish whether this species has a capacity for fatty acid oxidation.

## Exploiting differences in metabolism for potential pharmacological intervention

Differences in metabolism between *T. congolense* and *T. brucei* have implications for differential drug efficacy between the two species. To validate our findings in key areas of metabolism, pharmacological inhibition was attempted for specific targets in trypanosome metabolism, in order to compare inhibitory concentrations ($EC_{50}$).

To assess whether areas of mitochondrial metabolism were more necessary in BSF *T. congolense* than in BSF *T. brucei*, both species were treated with FCCP, an uncoupling agent that depolarises the mitochondrial membrane. However, there was no difference in sensitivity between the species ($EC_{50}$: $13.0 \pm 5.0$ μM and $12.6 \pm 5.3$ μM for *T. brucei* and *T congolense*, respectively; Table 1). Given both metabolic and transcriptomic data indicated no increased electron transport chain activity, we also treated with the complex III inhibitor antimycin A, again with no significant differences observed between the species (Table 1). In addition, there was no change in sensitivity to azide, an inhibitor of ATP hydrolysis by the $F_1$-ATPase (Table 1). However, *T. congolense* appeared to be less sensitive to rotenone, a complex I NADH dehydrogenase inhibitor (Table 1). Previous data inferred complex I activity in BSF *T. congolense* based on nitroblue tetrazolium staining [51]. Rotenone resistance could indicate NADH dehydrogenase activity of a rotenone-insensitive NADH dehydrogenase, such as the inner membrane space-facing NDH2 [77].

*T. congolense* also showed enhanced sensitivity to salicylhydroxamic acid (SHAM), an inhibitor of the trypanosome alternative oxidase (TAO; Table 1) [102]. Sensitivity of both species to SHAM was also tested in the presence of 10 mM glycerol (Table 1). Addition of glycerol as a carbon source has previously been shown to increase *T. brucei* sensitivity to SHAM [103].

**Table 1. Comparative analysis of sensitivity to metabolic inhibitors in *T. congolense* and *T. brucei*.** Abbreviations: FCCP, carbonyl cyanide-p-trifluoromethoxyphenyl-hydrazone; SHAM, salicylhydroxamic acid; TAO, trypanosome alternative oxidase; AceCS, acetyl-CoA synthetase.

| Compound | Target | *T. congolense* $EC_{50}$ Mean ± SEM | *T. brucei* $EC_{50}$ Mean ± SEM | Fold change (Tc/Tb) | P value (*t*-test) |
|---|---|---|---|---|---|
| Antimycin | Complex III | 271.2 ± 143.5 μM | 144.2 ± 18.1 μM | 1.9 | 0.4295 |
| FCCP | Uncoupling agent | 12.6 ± 5.3 μM | 13.0 ± 5.0 μM | 1.0 | 0.9592 |
| Azide | Complex V ($F_1$-ATPase) | 432.3 ± 127.9 μM | 235.0 ± 6.0 μM | 1.8 | 0.1982 |
| Oligomycin | Complex V ($F_0$-ATPase) | 33.9 ± 14.1 nM | 197.6 ± 39.0 nM | 0.2 | 0.0169 |
| Rotenone | Complex I | 27.4 ± 1.4 μM | 7.4 ± 0.9 μM | 3.7 | 0.0003 |
| SHAM | TAO | 30.22 ± 0.7 μM | 60.23 ± 1.8 μM | 2.0 | 0.0001 |
| SHAM + 10 mM glycerol | TAO | 2.28 ± 3.0 μM | 5.00 ± 0.1 μM | 2.2 | 0.0008 |
| UK5099 | Pyruvate transport | 82.1 ± 8.8 μM | 130.0 ± 5.0 μM | 0.6 | 0.0091 |
| AceCS inhibitor | Acetyl-CoA synthetase | 57.7 ± 15.2 μM | 7.1 ± 2.4 μM | 8.1 | 0.0304 |
| Orlistat | Fatty acid synthase/lipases | 15.6 ± 2.5 μM | 0.02 ± 0.01 μM | 780.0 | 0.0033 |
| Diminazene | Kinetoplast | 50.0 ± 5.6 nM | 32.0 ± 0.5 nM | 1.6 | 0.0425 |

Both species exhibited increased sensitivity to SHAM in the presence of glycerol ($EC_{50}$ of 2.28 ± 3.0 μM and 5.00 ± 0.1 μM for *T. congolense* and *T. brucei*, respectively). This represented a 12.1- and 13.3-fold increase in sensitivity for *T. congolense* and *T. brucei*, respectively. Taken together, these data indicate that, like *T. brucei*, *T. congolense* does not rely on oxidative phosphorylation for ATP production, as indicated by transcriptomics analysis, and that, as previously reported, TAO is the terminal oxidase [51,54].

Metabolomics and transcriptomics data indicated that *T. congolense* direct pyruvate towards mitochondrial metabolism, with high transcript levels in PDH and enzymes involved in acetate generation, compared to *T. brucei* (Figs 3 and 7). We therefore hypothesised that *T. congolense* would be more sensitive to inhibition of mitochondrial pyruvate uptake, and to investigate this further, we tested drug sensitivities for UK5099, an inhibitor of mitochondrial pyruvate transport [104]. As expected, *T. congolense* ($EC_{50}$: 82.1 μM) was significantly more sensitive (p = 0.0091, unpaired *t*-test) to UK5099 compared to *T. brucei* (130.0 μM; Table 1).

Whilst acetate generation appears to be important in *T. congolense*, our data suggest that the acetate does not appear to be utilised for the biosynthesis of fatty acids, in contrast to what has been shown for *T. brucei*. To probe this further, we compared drug sensitivity of the two species with compounds targeting fatty acid synthesis (S9 Fig). Indeed, *T. congolense* was significantly more resistant than *T. brucei* to an acetyl-CoA synthetase inhibitor (AceCS inhibitor; 1-(2,3-di(thiophen-2-yl)quinoxalin-6-yl)-3-(2-methoxyethyl)urea, [105]; S9 Fig and Table 1), indicating that AceCS is not as essential to this species. AceCS is essential to both BSF and PCF *T. brucei* [33,69], thus indicating a key metabolic difference between the species.

We next compared drug sensitivity to Orlistat, an inhibitor of fatty acid synthase and phospholipase [35]. Here, a striking difference was found, with *T. congolense* exhibiting significantly less sensitivity (780-fold increase in $EC_{50}$) to the compound compared to *T. brucei* (S9 Fig and Table 1), providing further evidence that *T. congolense* primarily relies on fatty acid scavenging, instead of synthesis, as predicted by the combination of metabolomics and transcriptomics.

## Discussion

The protozoan parasite *T. congolense* is a principal cause of AAT, but crucially, *T. brucei* remains the dominant model for laboratory-led studies of African trypanosomes, even in the face of mounting evidence that *T. brucei* and *T. congolense* differ profoundly in many facets of their biology. This study aimed to generate a detailed comparison of metabolism in *T.*

*congolense* and *T. brucei*, through a combination of metabolomics, transcriptomics and gene knockdown approaches. Based on these comparisons, areas of metabolism were further probed with chemical inhibition, in order to validate findings.

Transcriptomic data were generated from *T. congolense* and *T. brucei* with parasite samples isolated from both *in vitro* culture and *in vivo* murine infections (*ex vivo*). There were high levels of correlation between *ex vivo* and *in vitro T. congolense* samples, indicating that the cultured form of the parasite closely resembles the *in vivo* situation, at a transcriptomic level. In contrast, there was lower inter-species correlation between *T. brucei* and *T. congolense*.

To complement the transcriptomic data, several metabolomic analyses were carried out to gain an understanding of specific areas of metabolism. These data demonstrated that BSF *T. congolense*, while possessing some metabolic similarities with BSF *T. brucei*, differs substantially in several core components, including having a reduced reliance on glucose, excretion of distinct glycolytic end products (acetate, malate and succinate in *T. congolense* compared to pyruvate in *T. brucei*), and increased gene expression and metabolic signatures of specific mitochondrial pathways, in particular pyruvate to acetate conversion. Additionally, we have shown that *T. congolense* has increased reliance on exogenous substrates such as ribose for nucleotide synthesis, as demonstrated by reduced glucose-derived carbon labelling in nucleoside species in addition to upregulation of hydrolases and phosphoribosyltransferases. Furthermore, while there is overlap in amino acid utilisation (e.g. glutamine), *T. congolense* relies on more exogenous amino acids than *T. brucei*. Surprisingly, this included serine which, in the case of *T. congolense*, appears to be important in the transsulfuration pathway that is geared towards trypanothione biosynthesis. This may also explain the observed decreased reliance on exogenous cysteine. Unlike *T. brucei*, *T. congolense* also requires asparagine and proline for viable *in vitro* culture, although carbon usage from these amino acids is minimal. Finally, *T. congolense* exhibits increased acetate/acetyl-CoA metabolism compared to *T. brucei*, despite a reduction in fatty acid biosynthesis through the classical trypanosomatid pathways involving acetyl-CoA synthase, acetyl-CoA carboxylase, β-ketoacyl-CoA synthase and β-ketoacyl-CoA reductase, the expression of which are reduced in *T. congolense* (both in *ex vivo* and *in vitro* conditions). This is further underlined by lack of glucose- and threonine-derived carbon labelling of saturated fatty acids, most notably myristic acid, a key GPI anchor component of variant surface glycoproteins of *T. brucei* and *T. congolense* [98]. However, fatty acid elongase 4, previously shown to extend exogenously scavenged arachidonic acid (C22:4) to docosatetraenoic acid (C22:5) [106], was expressed at higher levels under *in vitro* conditions, compared to *T. brucei*, which may indicate a reliance on long-chain polyunsaturated fatty acids. These findings are summarised in Fig 8.

Analyses of culture supernatants showed that 10 mM glucose was not substantially depleted after *T. congolense* cultures reached high cell density, as would be expected from an equivalently dense *T. brucei* culture [60]. *T. brucei* requires at least 5 mM glucose in culture [71], whereas BSF *T. congolense* were viable and maintained doubling times in levels as low as 2 mM. This reduced flux indicates that *T. congolense* is unlikely to be as susceptible to glycolytic inhibition as *T. brucei*, where 50% inhibition is sufficient to kill the parasite [27]. Interestingly, we observed a reproducible reduction in pyruvate levels in *T. congolense* supernatants over time, before abundance of this metabolite returned to levels similar to those observed in negative controls. A recent study in PCF *T. brucei* demonstrated that these parasites can re-metabolize glycolytic end products such as pyruvate and succinate [40]. Stable isotope labelling patterns in catabolic products derived from glucose do not support cyclical TCA activity, nor re-uptake of excreted metabolites in BSF *T. congolense*. However, it would be of interest to determine whether this species can recycle the aforementioned metabolites.

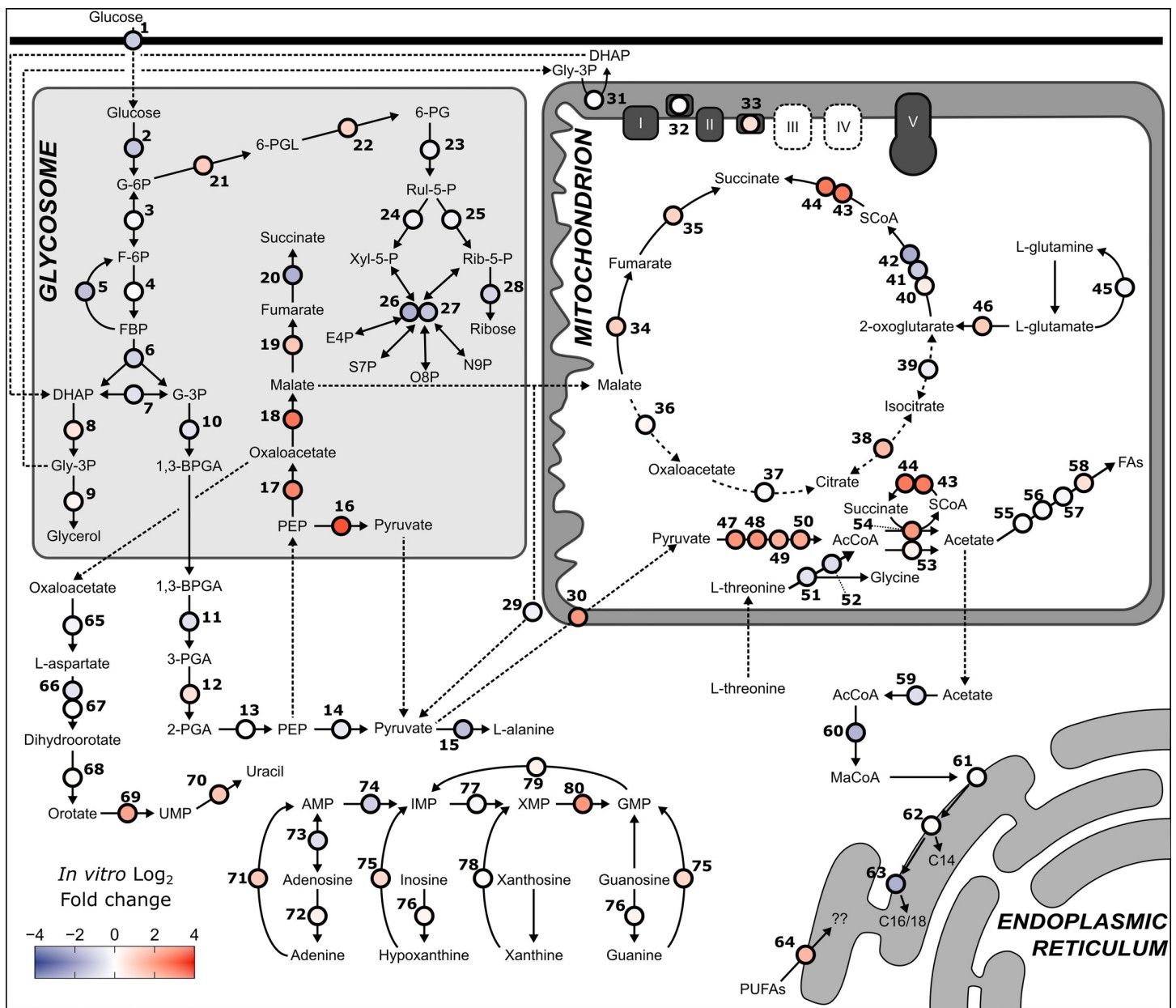

**Fig 8. Summary of *T. congolense* and *T. brucei in vitro* transcriptome.** $Log_2$ fold change (*T. congolense/T.brucei*) was calculated for each gene. Dashed lines represent transport processes. Genes: 1, hexose transporters, TbTc_0095; 2, hexokinase, TbTc_0341; 3, glucose-6-phosphate isomerase, TbTc_1840; 4, phosphofructokinase, TbTc_1399; 5, fructose-1,6-bisphosphatase, TbTc_1967; 6, aldolase, TbTc_0358; 7, triosephosphate isomerase, TbTc_1075; 8, glycerol-3-phosphate dehydrogenase, TbTc_2722; 9, glycerol kinase, TbTc_0392; 10, glyceraldehyde 3-phosphate dehydrogenase, TbTc_0377; 11, phosphoglycerate kinase, TbTc_0240; 12, phosphoglycerate mutase, TbTc_5039; 13, enolase, TbTc_0465; 14, pyruvate kinase 1, TbTc_0372; 15, alanine aminotransferase, TbTc_0675; 16, pyruvate phosphate dikinase, TbTc_1304; 17, Phosphoenolpyruvate carboxykinase, TbTc_0348; 18, glycosomal malate dehydrogenase, TbTc_0642; 19, glycosomal fumarate hydratase, TbTc_0242; 20, glycosomal NADH-dependent fumarate reductase, TbTc_0140; 21, glucose-6-phosphate dehydrogenase, TbTc_0931; 22, 6-phosphogluconolactonase, TbTc_4165; 23, 6-phosphogluconate dehydrogenase, TbTc_2025; 24, ribulose-5-phosphate epimerase, TbTc_4356; 25, ribose 5-phosphate isomerase, TbTc_3090; 26, transketolase, TbTc_1701; 27, transaldolase, TbTc_1823; 28, ribokinase, TbTc_5212; 29, malic enzyme, TbTc_0296; 30, Mitochondrial pyruvate carrier 2, TbTc_2668; 31, FAD-dependent glycerol-3-phosphate dehydrogenase, TbTc_2282; 32, NADH dehydrogenase (NDH2), TbTc_5033; 33, Alternative oxidase, TbTc_6589; 34, mitochondrial fumarate hydratase, TbTc_0243; 35, mitochondrial NADH-dependent fumarate reductase, TbTc_0141; 36, mitochondrial malate dehydrogenase, TbTc_0256; 37, citrate synthase, TbTc_0486; 38, aconitase, TbTc_5765; 39, isocitrate dehydrogenase, TbTc_0510; 40, 2-oxoglutarate dehydrogenase E1 component, TbTc_2864; 41, 2-oxoglutarate dehydrogenase E1 component, TbTc_3111; 42, 2-oxoglutarate dehydrogenase E2 component, TbTc_3057; 43, succinyl-CoA synthetase α, TbTc_0813; 44, succinyl-CoA ligase β, TbTc_3392; 45, glutamine synthetase, TbTc_2226; 46, glutamate dehydrogenase, TbTc_0872; 47, pyruvate dehydrogenase E1 α subunit, TbTc_4169; 48, pyruvate dehydrogenase E1 β subunit, TbTc_5437; 49, dihydrolipoamide acetyltransferase, TbTc_1015; 50, pyruvate dehydrogenase complex E3, TbTc_4765; 51, L-threonine 3-dehydrogenase, TbTc_5991; 52, 2-amino-3-ketobutyrate coenzyme A ligase, TbTc_6236; 53, Acetyl-CoA hydrolase (ACH), TbTc_5515; 54, Succinyl-CoA:3-ketoacid coenzyme A transferase (ASCT), TbTc_0236; 55, Acyl carrier protein, TbTc_5262; 56, beta-ketoacyl-ACP synthase, TbTc_3372; 57, beta-ketoacyl-ACP

reductase, TbTc_1241; 58, Trans-2-enoyl-ACP reductase 1, TbTc_5269; 59, acetyl-CoA synthetase, TbTc_0318; 60, acetyl-CoA carboxylase, TbTc_0754; 61, Fatty acid elongase (ELO1), TbTc_0159; 62, Fatty acid elongase (ELO2), TbTc_1882; 63, Fatty acid elongase (ELO3), TbTc_0235; 64, elongation of very long chain fatty acids protein (ELO4), TbTc_0737; 65, aspartate aminotransferase, TbTc_0799; 66, aspartate carbamoyltransferase, TbTc_1630; 67, dihydroorotase, TbTc_3801; 68, dihydroorotate dehydrogenase, TbTc_0620; 69, orotidine-5-phosphate decarboxylase/orotate phosphoribosyltransferase, TbTc_0735; 70, uracil phosphoribosyltransferase, TbTc_4220; 71, Adenine phosphoribosyltransferase (APRT-2), TbTc_3522; 72, inosine-adenosine-guanosine-nucleoside hydrolase, TbTc_4998; 73, adenosine kinase, TbTc_1024; 74, AMP deaminase, TbTc_5808; 75, hypoxanthine-guanine phosphoribosyltransferase (HGPRT), TbTc_0726; 76, inosine-guanine nucleoside hydrolase, TbTc_0808; 77, inosine-5'-monophosphate dehydrogenase, TbTc_1648; 78, Hypoxanthine-guanine-xanthine phosphoribosyltransferase (HGXPRT), TbTc_3696; 79, GMP reductase, TbTc_4627; 80, GMP synthase, TbTc_1452. Abbreviations: PUFA, polyunsaturated fatty acid.

*T. congolense* exhibits high levels of expression of genes in the glycosomal succinate shunt, with the exception of fumarate reductase, the last step of succinate production, for reasons we could not identify (Fig 8). In *T. brucei* these phenotypes are associated with PCF rather than BSF. Furthermore, whilst the major PGK isoform in BSF *T. brucei* is expressed in the glycosome, a previous study suggested that the major isoform of phosphoglycerate kinase in BSF *T. congolense* lacks the glycosomal targeting signal present in *T. brucei*, and is thus expressed in the cytosol, akin to PCF *T. brucei* [49]. This has significant implications for glycosomal ADP/ATP balance, as the expression of cytosolic PGK in BSF *T. brucei* is lethal [42]. Taken together, these data suggest that *T. congolense* appears to carry out glycolytic metabolism in the same fashion as PCF, not BSF *T. brucei*, including in *ex vivo* cells.

Whilst both 2-deoxy-D-glucose and GlcNAc have a detrimental impact on *T. congolense* viability, knock-down of the glucose transporter array did not affect growth, even though glucose uptake was reduced by 37% after 72 h of RNAi induction. Consistent with other studies RNAi penetrance does not appear as efficient as in *T. brucei* [107] and techniques such as CRISPR/Cas9 and conditional knock-out would greatly enhance our capabilities to study this parasite. Nevertheless, these experiments highlight a crucial difference between BSF *T. congolense* and *T. brucei* in a pathway that has become a metabolic paradigm in the latter species. Whilst *T. brucei* requires high levels of glucose to sustain a significant glycolytic flux, *T. congolense* remains viable in significantly lower glucose concentrations, with a reduced flux, again more similar to PCF *T. brucei*. However, glucose remains an essential carbon source in this species, as growth is abolished in the absence of glucose. Of particular interest is whether the parasite generates the majority of ATP from this reduced glucose intake, or if it can thrive on other carbon sources such as amino acids or even fatty acids. If the latter, this adaptation could be due to the reduced bioavailability of glucose in the ruminant host bloodstream. Blood concentrations of glucose in humans are approximately 5.5 mM [108]. Glucose concentrations in ruminants are typically lower (2–4 mM [109–111]), and primary sources of energy are typically volatile fatty acids in the form of acetic, propionic and butyric acid [112,113]. To date, products of volatile fatty acid metabolism, such as 2-methylcitrate and 2-methyl-cis-aconitase have not been reported in *T. congolense*. However, it is thought *T. brucei* can metabolise ketone bodies such as β-hydroxybutyrate through the action of a β-hydroxybutyrate dehydrogenase (Tb927.10.11930) to generate acetoacetate [114]. *T. congolense* possesses an orthologue of this gene (TcIL3000.A.H_000824100), and therefore, the ability of *T. congolense* to utilise other products available in adult ruminant blood merits further investigation.

RNAseq analyses of *T. congolense* indicate high levels of expression of mitochondrial pathways associated with glucose catabolism, specifically acetate and acetyl-CoA metabolism involving PDH, ASCT and succinyl-CoA synthetase (SCS; Fig 8). Given that the large amounts of acetate generated by the parasite appear not to be required for fatty acid synthesis, these findings could suggest significant reliance on mitochondrial substrate level phosphorylation for growth, similar to PCF *T. brucei* cultured in glucose-rich medium [115,116]. Interestingly, *T. congolense* does not appear to encode a syntenic orthologue of MPC1 (S1 Fig), and attempts

to identify the gene by orthoMCL or Orthofinder were unsuccessful. In *T. brucei*, the mitochondrial pyruvate carrier is composed of two paralogous proteins, MPC1 and MPC2, both of which are essential [104]. The absence of an MPC1 orthologue in *T. congolense*, in spite of these cells requiring mitochondrial pyruvate transport, inferred from acetate production from glucose, indicates a key difference in the constitution of pyruvate carrier in this species. Structural differences to pyruvate carriers between the two species, or the enhanced mitochondrial pyruvate catabolism in *T. congolense* may explain its enhanced sensitivity to UK5099, a mitochondrial pyruvate transport inhibitor. In addition, previous data suggest that in *T. brucei*, there is likely another mitochondrial pyruvate carrier that is insensitive to UK5099 [104].

Our data are consistent with the absence of oxidative phosphorylation, based on transcriptomics and lack of sensitivity to chemical inhibition, compared to *T. brucei*. One previous study reported NADH dehydrogenase activity in BSF *T. congolense*, which was presumed to originate from complex I, indicating potential ATP generation via complex V [51]. However, *T. congolense* exhibit significantly reduced sensitivity to rotenone, compared to *T. brucei*, suggesting that NADH dehydrogenase activity may originate from a rotenone-insensitive NADH dehydrogenase such as NDH2, known to be important for acetate production in BSF *T. brucei* [77,117,118]. This result must be treated with caution, as high concentrations of rotenone can lead to off-target effects [119]. Sensitivity to the TAO inhibitor SHAM suggests that TAO is the terminal oxidase, with no significant complex III or IV activity. These conclusions support one previous study of BSF *T. congolense* [51], although further work must be carried out to confirm the roles of complex I and V in this species.

Rather than oxidative phosphorylation, we propose it is more likely that considerable ATP production occurs in the ASCT–SCS cycle, which would explain the high levels of acetate generated by *T. congolense*, in addition to increased sensitivity to inhibition of mitochondrial uptake of pyruvate, the key metabolic precursor. Given that 2-oxoglutarate dehydrogenase complex expression appears to be less than, or equal to, that in *T. brucei* (under *in vitro* culturing conditions; Fig 8), it is likely that SCS activity occurs in the acetate-generating pathway rather than in the TCA cycle, which is not thought to be fully functional in BSF African trypanosomes [38], although recent data have challenged this paradigm in PCF *T. brucei* [40]. The mechanisms proposed here bear some similarities to the scheme proposed by Dewar and colleagues for stumpy-form *T. brucei* metabolism, which also exhibit increased mitochondrial metabolism compared to BSF *T. brucei* [120].

In *T. brucei*, carbon atoms from glucose disseminate through multiple pathways in the cell [45] and, using stable isotope-labelled glucose, our data demonstrate that this pattern is also seen in *T. congolense*, in particular through the glycolytic pathway, suggesting some of the key metabolic differences observed are quantitative, rather than qualitative. However, there were key differences in glucose-derived carbon usage. In particular, a reduction in labelling was observed in purine nucleotides in *T. congolense*. In both species, carbon labelling is likely due to generation of ribose phosphate sugars via the PPP and these data suggest that *T. congolense* does not obtain its ribose through the PPP (from glucose), to the same extent that *T. brucei* does. Interestingly, *T. congolense* appears to express higher levels of APRT1 (cytosolic) compared to APRT2 (glycosomal) to synthesise adenosine (Fig 5). This discrepancy could underpin the reduced fraction of glucose-derived purine labelling, with a reliance on ribose from alternative sources (for example, exogenously).

Whilst the majority of pyrimidine labelling is 5-carbons in *T. brucei*, indicating labelled ribose, there is decreased 5-carbon labelling and higher abundance of 2-carbon labelling in *T. congolense*, likely through uridine generated from aspartate through orotate, again highlighting a reduction in glucose-derived ribose, but conversely, an increase in glucose-derived UMP and its derivatives. This study was not able to assess whether *T. congolense* has a capacity for

cytosine uptake. It is established that *T. brucei* does not take up this pyrimidine [121], but given the expanded nucleobase transporter repertoire in the *T. congolense* genome [21], it would be of interest to carry out more in depth analysis of cytosine metabolism, as well as nucleotide metabolism in general, given the interest in drug development.

There was also a reduced abundance of glucose- and threonine-derived fatty acid labelling in *T. congolense* relative to *T. brucei*. One previous study noted that, like *T. brucei*, the major constituent of *T. congolense* GPI anchors is glycosyl-sn-1,2-dimyristylphosphatidylinositol, suggesting that this species will have a substantial requirement for myristic acid [98]. However, lack of glucose- or threonine-derived fatty acid labelling, in addition to decreased transcript abundance of AceCS, suggests that *T. congolense* may scavenge exogenous lipids in favour of carrying out fatty acid biosynthesis. Supernatant metabolomics showed accumulation of both choline and choline phosphate, with a corresponding decrease in putative LysoPCs, which appears to indicate activity of the phospholipases that *T. congolense* is known to secrete [62,122]. It is unknown whether *T. congolense* is able to generate cytosolic acetyl-CoA for fatty acid biosynthesis through the action of citrate lyase, although transcript abundance of this putative citrate lyase gene was reduced compared to *T. brucei*. Analysis of drug sensitivity supports the above conclusions, as *T. congolense* is significantly less sensitive to acetyl-CoA synthetase inhibition, as well as Orlistat, an inhibitor of fatty acid synthase, suggesting that fatty acid scavenging (e.g. lipid or fatty acid transporters) could be a viable therapeutic target for this species. However, the efficacy of orlistat *in vivo* has not been reported to our knowledge.

BSF *T. brucei* growth in CMM required only cysteine and glutamine when supplemented with FBS gold, although a further 6 amino acids (Tyr, Phe, Trp, Leu, Met and Arg) were required when supplemented with standard FBS [60]. As part of this study, 14 amino acids essential for *T. congolense* growth were identified. Tryptophan and arginine, essential to *T. brucei*, were not required to sustain *T. congolense* growth in 10% goat serum. Conversely, several amino acids considered not essential to *T. brucei* were crucial for *T. congolense* growth *in vitro* (Asp, His, Ile, Pro, Ser and Val). Proline is a well-established carbon source for PCF *T. brucei* [92]. However, based on stable isotope labelling experiments, this amino acid is solely used for protein synthesis in BSF *T. congolense*, as there was no evidence of carbon dissemination from proline into the metabolome (likewise for asparagine). Unlike BSF *T. congolense*, BSF *T. brucei* must be able to synthesise sufficient amounts of these amino acids from alternative sources, or obtain them from the serum supplement.

One metabolic area of interest in trypanosomatids is trypanothione biosynthesis, a crucial pathway for parasite response to oxidative stress. Indeed, trypanothione synthase, as well as proteins involved in the trypanothione biosynthesis pathway, such as ornithine decarboxylase (targeted by Eflornithine), have long been considered prime chemotherapeutic targets due to their absence from other organisms [123]. Whilst cysteine was previously known to be a main carbon contributor to trypanothione synthesis in *T. brucei* along with glutamine and methionine [89], we show here that serine, an amino acid essential to *T. congolense*, also contributes to the generation of this metabolite in addition to the aforementioned amino acids. These data indicate that *T. congolense* can both synthesise and transport cysteine. Interestingly, cysteine was not significantly depleted from *T. congolense* culture supernatants and future work should ascertain whether the presence of L-serine in medium can compensate for reduced cysteine levels in *T. congolense* culture.

The data presented here have led to the generation of a novel semi-defined medium for culturing the strain IL3000, which must be further optimized for the culture of multiple strains of *T. congolense*. Of interest is the peculiar requirement of adult bovine or goat serum for *in vitro* culture of *T. congolense*, rather than foetal bovine serum (FBS) which is typically used to culture *T. brucei* [18,63]. Whilst this study made no attempts to adapt *T. congolense* to FBS-

supplemented medium (indeed, even in SCM-7, growth rate is drastically reduced in the presence of FBS after 2–3 passages), this is of crucial importance, as it would allow the study of multiple species of African trypanosome under the same *in vitro* conditions. Analysis of metabolism presented here indicates that this phenomenon is likely to centre on the lipid requirements of *T. congolense*, although it remains to be seen if this requirement is for energy generation or synthesis of lipids in general. Furthermore, adult ruminant serum composition drastically differs from that of non-ruminants and of foetal ruminants [112,113], and this likely has significant implications on the extracellular environment faced by livestock trypanosomes.

The information presented here is a significant step in laying the foundation for fundamental understanding of metabolism for an important livestock parasite. Understanding essential areas of metabolism in both *T. brucei* and *T. congolense* enables the development of drugs effectively targeting both species. Conversely, understanding the key differences between the two species aids in dissecting drug mechanisms of action and resistance, as well as enabling a greater understanding of host-pathogen dynamics.

## Materials and methods

### Ethics statement

All animal experiments were performed in accordance with the Animals (Scientific Procedures) Act 1986 and the University of Glasgow care and maintenance guidelines. All animal protocols and procedures were approved by The Home Office of the UK government and the University of Glasgow Ethics Committee.

### Compounds and reagents

All compounds were obtained from Sigma/Merck with the exception of: Orlistat (Cambridge Bioscience), oligomycin A (VWR International), diminazene aceturate (Cambridge BioScience) and FCCP (Abcam).

### Cell lines and *in vitro* culture

In all cases, *T. congolense* strain IL3000 [51] was used (originally received from Theo Baltz, University of Bordeaux). For RNAi experiments a *T. congolense* IL3000 single marker line, TcoSM, was used [107]. For *in vitro* experiments, cells were grown at 34˚C, 5% $CO_2$ and routinely cultured in either TcBSF3 [17] or HMI-93 [18], in both cases without a serum plus supplement, with 20% goat serum (Gibco). For global metabolite analysis of culture supernatant, an experimental medium (Steketee's congolense medium-3; SCM-3) was used with the following components: 77 mM NaCl, 1.5 mM $CaCl_2$, 4.5 mM KCl, 0.8 mM $MgSO_4$, 36 mM $NaHCO_3$, 25 mM HEPES, 0.05 mM bathocuproinedisulfonic acid, 0.22 mM 2-mercaptoethanol, 50 U/mL penicillin/streptomycin, 2.5 mM glucose, 1 mM pyruvate, 10% goat serum, 10% TcBSF3 [17], 1 mM each of L-cysteine and L-glutamine, and 100 μM L-tyrosine, L-phenylalanine, L-tryptophan, L-leucine, L-methionine and L-arginine. BSF *T. congolense* in exponential growth phase were centrifuged at $1,500 \times g$ for 10 minutes, washed with PBS and inoculated into this medium (0 h time point).

For stable isotope labelling experiments, as well as experiments involving the removal or addition of specific medium components, a custom medium, Steketee's Congolense Medium-6 (SCM-6) was used (S5 Table). The final medium formulation based on this study's findings, SCM-7, is provided in S5 Table. This medium is essentially HMI-93, although i) vitamins (with the exception of folate) were removed, ii) D-glucose concentrations were modified depending on experimental procedure, but was routinely kept at 10 mM, iii) goat serum levels were reduced to

10% and, iv) of the 20 amino acids, 14 were added. Increasing the temperature to 37˚C led to a detrimental effect on cell viability after several passages, as previously reported [18].

For experiments involving *T. brucei*, either the monomorphic Lister 427 (*in vitro* experiments and growth curves) or pleomorphic STIB 247 (RNAseq experiments, both *in vitro* and *ex vivo* sample groups) strains were used. Lister 427 cells were grown in HMI-11 [124], whilst STIB 247 were grown in modified HMI-9 containing 1.1% methylcellulose and 20% serum plus (Sigma) [125,126]. In both cases, cells were incubated at 37˚C, 5% $CO_2$.

For both species, cell counts were carried out using a haemocytometer, and in the case of *T. congolense*, cells were mechanically detached from the culturing plasticware by pipetting prior to counting. Growth curves were routinely carried out in 2 mL samples incubated in 24-well plates, resuspended using a P1000. For detachment of cells from flasks, 10 mL plastic pipettes were used. In cases where cells were harvested for experiments other than those involving metabolomics, cells could also be detached by replacing the medium with PBS for incubating at room temperature for several minutes, prior to vigorously tapping the flask to detach parasites.

RNAi experiments using TcoSM were carried out in HMI-93 in 20 mL cultures. Cells were seeded at $7 \times 10^5$ cells/mL and RNAi induction was initiated with the addition of 1 μg/mL tetracycline (Sigma) and $1 \times 10^7$ cells were isolated every 24 hours for RNA analysis (outlined below) before cells were passaged.

## Animal experiments

Adult female CD-1 mice (20–30 g body weight; Charles River Laboratories) were infected with $5 \times 10^4$ wild-type *T. brucei* STIB 247 or $1 \times 10^5$ wild-type *T. congolense* IL3000 by intraperitoneal injection. Parasitaemia was monitored daily by venesection of the lateral tail vein [127]. At first peak of parasitaemia ($>10^7$ cells/mL) mice were euthanised and blood isolated. Parasites of both species were purified from blood by anion exchange using DEAE cellulose [128]. Purified cells were counted, and a total of $1 \times 10^8$ cells were centrifuged for 10 minutes at 1,500 $\times g$ prior to RNA extraction.

## RNA extraction

For *in vitro* parasite RNA isolation, samples ($10^8$ cells) of both *T. congolense* (IL3000) and *T. brucei* (STIB 247) were taken from actively dividing cultures grown to densities of 1.8–2 $\times 10^6$ cells/mL. For *ex vivo* parasite RNA isolation, samples (also $10^8$ cells) were derived from mouse infections as described above. RNA was extracted using the QIAgen RNeasy kit (Qiagen) with an on-column DNase treatment step. Sample concentrations were analysed by Nanodrop and QuBit, and concentrations adjusted to 37 ng/μL of which 80 μL (2.96 μg) was submitted for RNAseq.

For RNAi time course experiments, cell pellets ($10^7$ cells) were resuspended in 1 mL TRIzol (Invitrogen) and stored at -80˚C. Samples were thawed, 200 μL chloroform was added, samples were shaken vigorously for 15 seconds and incubated at room temperature for 3 minutes, prior to centrifugation at 12,000 $\times g$ for 15 minutes, 4˚C. The aqueous layer was transferred to a fresh tube and 500 μL isopropanol and 1 μL Glycoblue (Invitrogen) were added. Samples were mixed by inverting, incubated at room temperature for 10 minutes and centrifuged at 12,000 $\times g$ for 10 minutes at 4˚C. RNA pellet was washed in ice-cold 75% ethanol and centrifuged at 12,000 $\times g$ for 10 minutes at 4˚C. After air-drying, RNA was resuspended in 20 μL RNase-free water and concentration adjusted to 100 ng/μL. DNase treatment was carried out using the Ambion TURBO DNase kit (Applied Biosystems) as per manufacturer's instructions.

## Metabolomics sample preparation

For metabolomics analysis of supernatants, 10 mL *T. congolense* cultures were incubated in T25 flasks in relevant media. Cells were centrifuged at $1,500 \times g$ for 10 minutes, washed with PBS, resuspended in relevant media and density adjusted to $1 \times 10^5$ cells/mL. At each time-point, 500 μL medium was transferred to a 1.5 mL Eppendorf tube and briefly quenched in a dry ice/ethanol bath, before centrifuging at $1,500 \times g$ for 10 minutes at 4˚C. A 5 μL aliquot was then transferred to a new Eppendorf containing 200 μL metabolite extraction solvent (chloroform:methanol:water in a 1:3:1 ratio) and samples vortexed at 4˚C for one hour. Samples were centrifuged for 5 minutes at $13,000 \times g$ (4˚C) and supernatants transferred to new Eppendorf tubes. Samples were stored at -80˚C prior to analysis.

For analysis of intracellular metabolites, cells were grown to a final density of $2 \times 10^6$ cells/mL and a total of $10^8$ cells isolated. Cells were quenched in 50 mL falcon tubes to 4˚C using a dry ice/ethanol bath (stirred and measured by thermometer) and all subsequent steps were carried out at 4˚C. Cells were centrifuged at $1,500 \times g$ for 10 minutes and if supernatant samples were required in addition to cell pellets, 5 μL was transferred to an Eppendorf containing 200 μL extraction solvent. Cells were resuspended in residual medium before transfer to Eppendorf tubes. Cells were then centrifuged ($1,500 \times g$, 5 minutes) and washed twice with ice-cold phosphate buffered saline (PBS) before resuspension in 200 μL extraction solvent (chloroform:methanol:water in a 1:3:1 ratio). Samples were vortexed at 4˚C for 1 hour, and then centrifuged for 5 minutes at $13,000 \times g$. Supernatants were transferred to clean Eppendorf tubes. For all experiments, a quality control sample was generated by pooling 10 μL from each sample and samples were stored under argon gas at -80˚C.

## Primers and plasmids

RNAi experiments were carried out using a *T. congolense* single marker line, TcoSM [107] that expresses Tet repressor and T7 polymerase, maintained in 0.5 μg/mL puromycin, and gene specific RNAi constructs were introduced with a *T. congolense* specific plasmid, p3T7-TcoV [107]. Primers carrying a HindIII (forward) or an FseI (reverse) restriction site were used to amplify a fragment of *TcoHT* (fwd: AAGCTTAAACAGAGCAATGCCAGTCG; rev: GGCCGGCCTTATTACGTTTGGCATTATG; restriction sites underlined). Gene fragments were amplified using a HiFi polymerase master mix (NEB) and cloned into pGEM-T easy (Promega) and sequenced to confirm correct sequence identity of each fragment. The constructs were then digested with HindIII and FseI and ligated into the p3T7-TcoV vector [107] using T4 DNA ligase (Promega). The final plasmid was linearised with NotI before purification by ethanol precipitation prior to electroporation into TcoSM cells.

## Transfections/electroporations

*T. congolense* IL3000 electroporation experiments and selection experiments were performed as developed by [107]. A total of $4 \times 10^7$ cells were used per transfection, including a negative (buffer only) control. A transfection buffer previously published for use with *T. brucei* was used for *T. congolense* transfections [129]. Cells were centrifuged at $1,500 \times g$ for 10 minutes, pellets resuspended in residual medium and transferred to Eppendorf tubes for a further centrifugation step. Cells were subsequently washed in transfection buffer prior to final resuspension in 100 μL buffer per transfection. Up to 12 μg linearised plasmid DNA was added to an electroporation cuvette (Sigma), and 100 μL cells were subsequently added. Electroporation was carried out using a Nucleofector II (Lonza) programme Z-001. Transfected cells were then incubated overnight in 25 mL warm medium in the absence of selective antibiotics, prior to their addition and plating out at dilutions of 1:50, 1:100 and 1:200 in 96-well plates. Antibiotics

were added at the following concentrations: Puromycin: 0.5 μg/mL; Neomycin (G418): 0.4 μg/mL. Clones were retrieved after 7–10 days, and these were maintained in 0.25 μg/mL puromycin and 0.2 μg/mL G418.

## Drug sensitivity assays

Drug sensitivity assays were carried out using the alamar blue method developed by Raz and colleagues [130]. Briefly, Compounds were diluted to 2× starting concentration in SCM-6 (with 10% goat serum for *T. congolense* IL3000 or 10% FBS for *T. brucei* Lister 427) and 200 μL was transferred to the first well of a solid white flat-bottomed 96-well plate. 100 μL medium was then added to 23 further wells and compounds were diluted 1:2 over this series of wells, with the exception of the last well, for a negative control. Subsequently, 100 μL cells were added at 2× starting density ($4 \times 10^4$ cells/mL for *T. brucei* and $5 \times 10^5$ cells/mL for *T. congolense*). Plates were incubated for 48 hours (37˚C or 34˚c for *T. brucei* and *T. congolense*, respectively, 5% $CO_2$ in both cases), prior to addition of 20 μL resazurin sodium salt (0.49 mM in 1× PBS, pH 7.4) to each well. Plates were then incubated for a further 24 hours before measurements of cell viability.

Reduction of the resazurin salt was measured as a function of cell viability. Fluorescence of each plate was read using a Cytation 5 imaging reader (BioTek) and GEN5 software. Parameters were as follows: $\lambda_{excitation}$ = 540 nm and $\lambda_{emission}$ = 590 nm. Raw values were plotted against concentrations (converted to $\log_{10}$ values) and normalised (0% defined as smallest mean in the data; 100% defined as largest mean in the data) using Graphpad Prism version 8.4.0. $EC_{50}$ values for each compound were calculated using a non-linear sigmoidal dose-response curve. Each assay was performed in duplicate and each $EC_{50}$ value represents a mean of three independent experiments.

## Real-time quantitative PCR (RT-qPCR)

RNA was extracted as described above, and reverse transcription was carried out in 20 μL using 1 μg RNA, using a high capacity cDNA kit (Applied Biosystems). Primers for RT-qPCR analysis were designed using Primer 3 [131], and primer efficiency was tested using serial dilutions of *T. congolense* IL3000 genomic DNA by plotting Ct value against $\log_{10}$(DNA concentration). Real-time PCR was carried out using the SensiFAST SYBR Hi-ROX kit (Bioline, BIO92005). Briefly, a 20 μL reaction was set up using 10 μL SYBR mix, RT template and 400 nM of each primer. Primers targeting all five HT genes were used (TcoHT_fwd: ATAGT GACGGAGCGGTTCAT; TcoHT_rev: GACGCAACGACACCAATGAT). Cycling conditions were: 96˚C, 120 seconds, followed by 40 cycles of 95˚C for 5 seconds, 62˚C for 10 seconds and 72˚C for 20 seconds. Previously published endogenous control primers for TcoTERT were used for within sample normalisation (TcoTert_fwd: TTTCGCCCTCGTTTTCCTCA; TcoTert_rev: AGAAATCACGACCACACGCT) [132], and normalised transcript level was calculated using the delta delta Ct method [133].

## Glucose uptake assays

For analysis of wild-type *T. congolense* and *T. brucei* glucose uptake, cells were seeded in 10 mL cultures of SCM-6 at an initial density of $2 \times 10^5$ cells/mL (four cultures per species), with 10 mM glucose added separately at the start of the experiment. Upon the addition of glucose, 1 mL supernatant was immediately centrifuged ($1,500 \times g$, 10 minutes) and supernatant stored at -80˚C. This process was repeated at 12, 15, 18, 21 and 24 h, and cell density measured by haemocytometer. A medium-only control (4 replicates) was also incubated alongside *in vitro* cultures. Glucose concentration of each supernatant sample was analysed using the Glucose

(GO) assay kit (GAGO-20; Sigma) in a 96-well format. Briefly, 40 μL supernatant sample (diluted if necessary) was incubated with 80 μL assay reagent for 30 minutes at 37°C, after which 80 μL 12 N sulphuric acid was added and absorbance measured at 540 nm using a spectrophotometer. A standard curve was also run to calculate glucose concentration. Rate of glucose consumption was calculated using a custom script [61]. Doubling times of *T. congolense* IL3000 (11–12 hours) and *T. brucei* Lister 427 (6–7 hours) were taken into account when calculating rate of glucose consumption.

For glucose consumption of the TcoHT RNAi line, the Glucose Uptake-Glo kit (Promega) was used. RNAi was induced for 72 hours prior to carrying out the assay. Cells were centrifuged, washed in PBS and resuspended in assay buffer (77 mM NaCl, 1.5 mM $CaCl_2$-$2H_2O$, 4.5 mM KCl, 0.8 mM $MgSO_4$-$7H_2O$, 36 mM $NaHCO_3$, 25 mM HEPES and 0.02 mM bathocuproinedisulfonic acid), as it was determined *T. congolense* viability is reduced in PBS alone. Density was adjusted to $10^8$ cells/mL, and three 100 μL replicates of each sample were added to wells of a black flat-bottomed 96-well plate. The uptake reaction was started by the addition of 50 μL 1 mM 2-deoxy-D-glucose. Plate was shaken for 15 minutes at 34°C prior to addition of 25 μL stop buffer, 25 μL neutralisation buffer and 100 μL pre-prepared 2DG6P detection reagent. Plates were shaken in between addition of the buffers. Finally, the plate was read with 0.3–1 second integration on a luminometer (Cytation 5 Imaging reader, BioTek). Wild-type *T. congolense*, and *T. congolense* supplemented with glucose were used as controls, in addition to cells without 2-deoxy-D-glucose and assays in the absence of cells.

### Acetate concentration assay

Acetate was not detectable by mass spectrometry and therefore, a commercial colorimetric acetate assay kit (MAK086, Merck) was used to analyse changes in supernatant acetate concentrations in trypanosome cultures over time. *T. congolense* IL3000 were seeded in 10 mL SCM-6 at a density of $1 \times 10^5$ cells/mL as outlined in the supernatant metabolomics experiment. At each time-point, 500 μL supernatant was taken from each flask and transferred to an Eppendorf tube. Samples were centrifuged at $1,500 \times g$, the supernatant was transferred to a fresh Eppendorf tube, and samples were stored at -80°C until samples from all time-points had been collected. Acetate concentration assays were carried out according to the manufacturer's instructions. Briefly, for each sample, 5 μL was added to the wells of a 96-well plate in duplicate and 45 μL assay buffer was added to each sample. Subsequently, 50 μL reaction mix was added to each well, and the plate was mixed and incubated for 40 minutes at room temperature before absorbance was read at 450 nm. Acetate concentrations were calculated using the standard curve comprised of six concentrations run alongside the experimental samples.

### Metabolomics–liquid chromatography mass spectrometry

Hydrophilic interaction liquid chromatography (HILIC) was carried out by Glasgow Polyomics (Glasgow, UK), using a Dionex UltiMate 3000 RSLC system (Thermo Fischer Scientific) coupled to a ZIC-pHILIC column (150 mm × 4.6 mm, 5 μm column, Merch Sequant). The column was maintained at 30°C and samples were eluted with a linear gradient (20 mM ammonium carbonate in water and acetonitrile) over 26 minutes with a flow rate of 0.3 mL/minute.

Sample injection volume was 10 μL and samples were maintained at 4°C before injection. A Thermo Orbitrap Exactive (Thermo Fischer Scientific) was used to generate mass spectra, and was operated in polarity switching mode with the following settings: Resolution: 50,000; AGC: 106; m/z range: 70–1,400; sheath gas: 40; auxiliary gas: 5; sweep gas: 1; probe temperature: 150°C; capillary temperature: 275°C. Samples were run in both positive and negative polarity with the following ionisation: source voltage +4.5 kV, capillary voltage +50 V, tube voltage +70

kV and skimmer voltage +20 V for positive mode; source voltage -3.5 kV, capillary voltage -50 V, tube voltage -70 V and skimmer voltage -20 V for negative mode. Mass calibration was performed for each polarity immediately prior to each analysis batch. The calibration mass range was extended to cover small metabolites by inclusion of low-mass contaminants with the standard Thermo calmix masses (below $m/z$ 1400), $C_2H_6NO_2$ for positive ion electrospray ionisation (PIESI) mode ($m/z$ 76.0393) and $C_3H_5O_3$ for negative ion electrospray ionisation (NIESI) mode ($m/z$ 89.0244). To enhance calibration stability, lock-mass correction was also applied to each analytical run using these ubiquitous low-mass contaminants. A set of authentic standards was run prior to the sample set for each experiment.

## Metabolomics data analysis

RAW spectra were converted to mzXML files (mzML files for fragmentation data) using XCMS for untargeted peak detection [134]. The resultant files were further processed using mzMatch [135] for peak matching and annotation, resulting in a tabular output that was analysed using IDEOM with default settings [136]. Identification of metabolites was performed in Ideom (as previously described by [45]), by matching accurate masses and retention times of authentic standards (MSI confidence level 1). When standards were not available, predicted mass and retention time was calculated by a validated model [137] (MSI confidence level 2; putative annotation based on exact mass determination). Many of the metabolites names given in the datasets in this study are generated automatically as the Ideom software provides a best match to public database entries of the given mass and formula. Published literature and pathway/genome databases were considered to improve annotation in cases where isomers could not be differentiated based on accurate mass and retention time. Metabolites included in the manuscript were manually annotated using authentic standards where available. However, note that many of the metabolite names given in the IDEOM file are generated automatically as the software provides a best match to database entries of the given mass and formula (S3 Table). In the absence of additional information these must be considered as putatively-annotated hits; the confidence score in the column adjacent to putative metabolite name (S3 Table) serves as a guide to this. In addition, potential isomers matching the formula are provided in a drop down list for each metabolite (S3 Table). Clearly it is beyond the scope of any study to provide authenticated annotations to many hundreds of detected compounds, but the full datasets are included in the spirit of open access data.

For stable-isotope assisted metabolomics experiments, mzMatch output (in.peakml format) was analysed using mzMatch-ISO to extract all carbon isotopologue abundances from putative metabolites [138]. Data analysis of stable isotope-labelled metabolomics was based on a 48 hour time-point in all experiments. Data was further analysed using Microsoft Excel, R or Metaboanalyst v4.0 [139]. The raw data from all metabolomics analyses are available in Metabolights (accession number: MTBLS2372; URL: www.ebi.ac.uk/metabolights/MTBLS2372).

## RNA sequencing and data processing

RNA sequencing was carried out by Edinburgh Genomics (Edinburgh, UK). Libraries were prepared from 16 samples (8× *T. brucei*, 8× *T. congolense*) using the TruSeq Stranded mRNA kit (Illumina) and 2 × 75 bp paired-end sequencing was carried out using a HiSeq 4000 system (Illumina). Sequencing reads were aligned to the corresponding genome sequence using HiSat2 (default settings with "*-k 1—no-spliced-alignment*" to limit multi-mapping reads to one alignment) [140]. For *T. brucei*, the TREU 927 reference genome sequence was used (v51.0 from TriTrypDB [141]), whilst a PacBio assembly of *T. congolense* IL3000 (2019, v51.0 from TriTrypDB) was used for *T. congolense* [142]. The resulting SAM files were converted to BAM

files using samtools [143], and subsequently filtered for quality (*-q 1*). Read counts were extracted from the filtered BAM files using HTSeq-count ("*-s reverse -f bam -t CDS -i Parent -m union -a 0—nonunique-all*"). "CDS" was chosen as feature for read counting instead of "exon", as UTRs are not annotated in the *T. congolense* genome, in contrast to the *T. brucei* TREU 927 genome, which could lead to further discrepancies during cross-species transcriptome analysis. Read counts are provided in S1 Table.

For all samples, transcripts per million (TPM) values for each gene were calculated manually from the read count files using Microsoft Excel as follows: 1) Reads per kilobase (RPK) were calculated by dividing the read counts by the length of gene in kilobases (S1 Table); 2) All RPK values in a sample were summed and divided by 1 million as a scaling factor (S1 Table); 3) Each RPK value was divided by the scaling factor to yield TPM values [56]. To compare transcript abundances between the two species, Orthofinder [57] was used to infer orthologue genes or gene groups. Default parameters were used to compare the *T. brucei* TREU 927 and the *T. congolense* IL3000 annotated proteins (S2 Table). A custom MATLAB (version R2020a) script was used to combine the Orthofinder dataset and the TPM values for 1-to-1 orthologues, as well as "sum of TPM" values for groups containing multiple genes, where TPM value for each gene was summed, resulting in a final dataset (S1 Table). Raw RNA-seq data is deposited at GEO (accession number: GSE165290). Transcriptomics data were cross-referenced with the TrypanoCyc database (vm-trypanocyc.toulouse.inra.fr/; [64]) to enable pathway analysis of the data.

Statistical analysis of the orthoTPM dataset was performed using the limma package in R [144]. Briefly, genes with an orthoTPM value of < 1 across three or more of the four sample groups were removed from the dataset (98 genes). The data set was then log-transformed ($\log_2[\text{TPM}+1]$) prior to analysis with the eBayes() function including the parameters "trend = TRUE" and "robust = TRUE". Statistical output is included in S1 Table.

## Computation

Figures were generated using Graphpad Prism version 8.4.0 (www.graphpad.com) with the exception of scatter plots and heatmaps, which were generated using R [145]. Heatmaps were generated using the R packages pheatmap and ComplexHeatmap [146]; scatter plots were generated using GGplot2 and GGally; synteny plots were generated using the python tool MCScan [147]; pathway maps were generated with Inkscape v1.0.

## Supporting information

**S1 Fig. Synteny analysis of *T. brucei* and *T. congolense*.** Plots were generated to assess levels of synteny between the species in regions where genes appear to be absent in *T. congolense*. For 3 genes, SDH11 (top), MPC1 (middle) and a putative delta-4 fatty acid desaturase (bottom), surrounding regions are highly syntenic between *T. brucei* and *T. congolense*, indicating that these are likely 3 deletions from the *T. congolense* genome. Whilst this does not rule out existence of these genes in other genomic regions, approaches such as orthoMCL and BLAST did not yield high probability orthologues in *T. congolense*.
(TIFF)

**S2 Fig. Comparative analysis of published *T. congolense* RNAseq data and data generated in this study.** Scatter matrix of *T. congolense* datasets from this study compared to ascending and peak parasitaemia *in vivo* transcriptomics data generated by Silvester and colleagues [59]. TPM values were calculated for each gene in the *T. congolense* genome (S1 Table) and $\log_2(\text{TPM}+1)$ was plotted. Lower panels: Scatter plots of individual comparisons of the 4

datasets. Red dots correspond to genes associated with glycolysis, green dots correspond to genes possessing transmembrane domains that are likely to be transporters; Diagonal panels: sample names; Upper panels: Pearson correlation coefficients for comparisons of entire datasets (black), glycolytic pathway ("Glyc", green) and proteins with predicted transmembrane domains ("Trans", red).
(TIFF)

**S3 Fig. Comparative transcriptomics analysis of the electron transport chain in *T. congolense* and *T. brucei*.** A heatmap of all ETC complexes based on a table generated by Zikova and colleagues [76]. Heatmaps are divided into the alternative oxidases (AOX), NADH dehydrogenase 2 (NDH2), complex I, II, III, IV and ATPase (complex V).
(TIFF)

**S4 Fig. Stable isotope labelled ($^{13}$C)-glucose derived pyrimidine labelling.** Comparative analysis of glucose-derived pyrimidine labelling in *T. congolense* and *T. brucei* (taken from [45]).
(TIFF)

**S5 Fig. Effect of cysteine exclusion on *T. congolense* growth.** Parasites were grown in SCM-6 supplemented with 1.5 mM, 1.0 mM or absence of L-cysteine. Cell density was monitored every 24 hours.
(TIFF)

**S6 Fig. Comparison of amino acid metabolism in *T. congolense* and *T. brucei*.** A) glucose-derived carbon labelling of amino acids B) Transcriptomics pathway analysis. TrypanoCyc pathways and gene IDs: A) Arginine and polyamine synthesis (ARG+POLYAMINE-SYN): SpSyn, spermidine synthase, TbTc_1034; AdoMetDC_3193, AdoMet decarboxylase, TbTc_3193; ODC, ornithine decarboxylase, TbTc_5903; AdoMetDC_0696, AdoMet decarboxylase, TbTc_0696. B) Aspartate and asparagine biosynthesis (ASPASN-PWY): ASNS, asparagine synthetase, TbTc_4894; mASAT, mitochondrial aspartate aminotransferase, TbTc_5877; cASAT, cytosolic aspartate aminotransferase, TbTc_0799. C) Glutamate degradation (GLUCAT-PWY): OGDH-E1, 2-oxoglutarate dehydrogenase E1, TbTc_2864; GDH, glutamate dehydrogenase, TbTc_0872; SUCLG2, succinyl-CoA ligase, TbTc_3392; SCSα, succinyl-CoA synthetase, TbTc_0813; OGDH-E2, 2-oxoglutarate dehydrogenase E2, TbTc_3057. D) Isoleucine degradation (ILEUDEG-PWY): ECH, enoyl-CoA hydratase, TbTc_3283; BCAAT, branched-chain amino acid aminotransferase, TbTc_0559; SCP2, 3-ketoacyl-CoA thiolase, TbTc_4024. E) Leucine degradation (LEUDEG-PWY): ECH, enoyl-CoA hydratase, TbTc_3283; BCKDHα, 2-oxoisovalerate dehydrogenase α, TbTc_1182; AUH, methylglutaconyl-CoA hydratase, TbTc_5348; BCKDHβ, 2-oxoisovalerate dehydrogenase β, TbTc_0682; MCCα, 3-methylcrotonyl-CoA carboxylase α, TbTc_1670; IVDH, isovaleryl-CoA dehydrogenase, TbTc_3112; SCP2, 3-ketoacyl-CoA thiolase, TbTc_4024; HMGCL, hydroxymethylglutaryl-CoA lyase, TbTc_6160; BCAAT, branched-chain amino acid aminotransferase, TbTc_0559; MCCβ, 3-methylcrotonyl-CoA carboxylase β, TbTc_5385. F) Aspartate superpathway (PWY0-781): NADSYN, NAD+ synthase, TbTc_2404; mASAT, mitochondrial aspartate aminotransferase, TbTc_5877; METK1, AdoMet synthase, TbTc_0178; NMNAT, nicotinamide/nicotinic acid mononucleotide adenylyltransferase, TbTc_4133; cASAT, cytosolic aspartate aminotransferase, TbTc_0799; MTR, 5-methyltetrahydropteroyltriglutamate-homocysteine S-methyltransferase, TbTc_5805. G) Threonine degradation (PWY1V8-11): AKCT, 2-amino-3-ketobutyrate-CoA ligase, TbTc_6236; TDH, L-threonine dehydrogenase, TbTc_5991. H) Valine degradation (VALDEG-PWY): ECH, enoyl-CoA hydratase, TbTc_3283; HOPR, 2-hydroxy-3-oxopropionate reductase, TbTc_2903; BCAAT, branched-

chain amino acid aminotransferase, TbTc_0559. I) Proline degradation (PROLINE--DEG2-PWY): ProDH, proline dehydrogenase, TbTc_1591; GDH, glutamate dehydrogenase, TbTc_0872; P5CDH, delta-1-pyrroline-5-carboxylate dehydrogenase, TbTc1695.
(TIFF)

**S7 Fig. Carbon utilisation for trypanothione biosynthesis in *T. congolense*.** Metabolomics and transcriptomics analyses were carried out to analyse trypanothione biosynthesis. A) A simplified map of trypanothione biosynthesis as known in *T. brucei*. Numbers refer to the following enzymes: 1, S-adenosyl-L-methionine synthase, METK1; 2, S-adenosyl-L-methionine decarboxylase, AdoMetDC; 3, spermidine synthase, SpSyn; 4, methyltransferase reaction, MTase; 5, S-adenosyl-L-homocysteine dehydrolase, AdoHycase; 6, cystathionine beta synthase, CBS; 7, cystathione gamma lyase, CTH; 8, glutaminase/amidase, AM; 9, gamma-glutamylcysteine synthetase, GCS; 10, glutathione synthetase, GSS; 11, ornithine decarboxylase, ODC; 12, spermidine synthase, SpSyn; 13, glutathionylspermidine synthase, GSP; 14, trypanothione synthetase, TRYS; 15, tryparedoxin peroxidase, TXN1b; 16, trypanothione reductase, TRYR. B) Isotopologue labelling experiments using 100% $^{13}$C-L-serine, $^{13}$C-L-glutamine, $^{13}$C-L-methionine or $^{13}$C-L-cysteine, showing the abundance of carbon labelling derived from these amino acids in components of the trypanothione biosynthesis pathway. C) Transcriptomics analysis. TrypanoCyc pathways and gene IDs: Trypanothione biosynthesis (PWY1V8-6): AdoMetDC_0696, S-adenosylmethionine decarboxylase, TbTc_0696; GSS, glutathione synthetase, TbTc_3678; AM, amidase, TbTc_5549; AdoMetDC_3193, S-adenosylmethionine decarboxylase, TbTc_3193; TRYS, trypanothione synthetase, TbTc_1359; ODC, ornithine decarboxylase, TbTc_5903; TRYR, trypanothione reductase, TbTc_4239; SpSyn, Spermidine synthase, TbTc_1034; TNX1b, tryparedoxin 1b, TbTc_0324; METK1, S-adenosylmethionine synthetase, TbTc_0178; GCS, gamma-glutamylcysteine synthetase, TbTc_3424. Homocysteine degradation/cysteine biosynthesis (HOMOCYSDESGR-PWY1): CBS, cystathionine beta synthase, TbTc_0413; CTH, cystathione gamma lyase, TbTc_1051. Methionine degradation I (METHIONINE-DEG1-PWY): AdoHcyase, S-adenosylhomocysteine hydrolase, TbTc_0685; METK1, S-adenosylmethionine synthase, TbTc_0178.
(TIFF)

**S8 Fig. Analysis of LC-MS utilising stable isotope labelled amino acids.** Percentage total labelling of metabolites identified in data from 6 stable isotope labelling experiments using $^{13}$C-L-asparagine, $^{13}$C-L-cysteine, $^{13}$C-L-glutamine, $^{13}$C-L-methionine, $^{13}$C-L-proline and $^{13}$C-L-serine. Colour intensity correlates to the total fraction of the metabolite that was $^{13}$C-labeled.
(TIFF)

**S9 Fig. Inhibition of fatty acid synthesis in *T. brucei* and *T. congolense*.** Sigmoidal dose-response curves to determine differential sensitivity of the two species of parasite to inhibition of an ACS inhibitor (panel A) and Orlistat (B).
(TIFF)

**S1 Table. RNAseq dataset–*T. congolense ex vivo*, *T. congolense in vitro*, *T. brucei ex vivo*, *T. brucei in vitro*, Silvester *et al*. [59] dataset.** Separate worksheets include the final OrthoTPM dataset, raw read counts (HTSeq-count output), RPK counts, scaling factors and TPM counts for each sample aligned to its respective genome. A legend is provided in the first worksheet.
(XLSX)

**S2 Table. Orthofinder output comparing *T. congolense* IL3000, *T. congolense* IL3000 2019, *T. brucei* TREU 927 and other trypanosomatids.** Further worksheets include lists of genes

present in only *T. brucei* or only *T. congolense*, with no orthologue detected in the other.
(XLSX)

**S3 Table. Supernatant metabolomics dataset for *in vitro* cultured *T. congolense* over a period of 56 hours. Metabolites highlighted in yellow were confidently predicted (MSI level 1) using a set of metabolite standards run alongside the experimental samples.** Where applicable, a list of potential isomers based on matching formula is provided. Note that confidence value only applies to the original metabolite identified by the IDEOM software. Results of statistical analysis by means of a one-way repeated measures ANOVA (false discovery rate-adjusted P value, FDR) is also shown for metabolites that were taken forward for downstream analysis. Further details for each column are provided in a separate worksheet in the same excel file.
(XLSX)

**S4 Table. TrypanoCyc pathways and linked Orthogroup (TbTc) gene IDs.**
(XLSX)

**S5 Table. Formulation of Steketee's congolense medium (SCM)-6 & -7.**
(XLSX)

## Acknowledgments

The authors would like to thank the following people: Steve Kelly, Paul Michels & Simon Young for very useful discussion. We thank Anne-Marie Donachie for help with *in vivo* experiments. In addition, the authors would like to thank the following institutes: Glasgow Polyomics for LC-MS work, Edinburgh Genomics for RNAseq, MRC PPU DNA Sequencing and Services, Dundee, for plasmid sequencing.

## Author Contributions

**Conceptualization:** Kathryn Crouch, Harry P. de Koning, Catarina Gadelha, Michael P. Barrett, Liam J. Morrison.

**Data curation:** Pieter C. Steketee, Emily A. Dickie, James Iremonger, Edith Paxton, Siddharth Jayaraman, Omar A. Alfituri, Ryan Ritchie.

**Formal analysis:** Pieter C. Steketee, Emily A. Dickie, James Iremonger, Kathryn Crouch, Edith Paxton, Siddharth Jayaraman, Omar A. Alfituri, Ryan Ritchie, Liam J. Morrison.

**Funding acquisition:** Harry P. de Koning, Michael P. Barrett, Liam J. Morrison.

**Investigation:** Pieter C. Steketee, Emily A. Dickie, Harry P. de Koning, Michael P. Barrett, Liam J. Morrison.

**Methodology:** Pieter C. Steketee, James Iremonger, Kathryn Crouch, Siddharth Jayaraman, Georgina Awuah-Mensah, Catarina Gadelha, Bill Wickstead, Michael P. Barrett, Liam J. Morrison.

**Project administration:** Tim Rowan, Michael P. Barrett, Liam J. Morrison.

**Resources:** Michael P. Barrett, Liam J. Morrison.

**Supervision:** Achim Schnaufer, Liam J. Morrison.

**Validation:** Pieter C. Steketee, Bill Wickstead, Michael P. Barrett.

**Visualization:** Pieter C. Steketee.

**Writing – original draft:** Pieter C. Steketee, Emily A. Dickie, Liam J. Morrison.

**Writing – review & editing:** Pieter C. Steketee, Emily A. Dickie, Georgina Awuah-Mensah, Achim Schnaufer, Tim Rowan, Harry P. de Koning, Catarina Gadelha, Bill Wickstead, Michael P. Barrett, Liam J. Morrison.

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
