## [Decision Letter · Decision Letter 0]

23 Mar 2021

Dear Dr Steketee,

Thank you very much for submitting your manuscript "Divergent metabolism between Trypanosoma congolense and Trypanosoma brucei results in differential drug sensitivity" for consideration at PLOS Pathogens. As with all papers reviewed by the journal, your manuscript was reviewed by members of the editorial board and by several independent reviewers. In light of the reviews (below this email), we would like to invite the resubmission of a significantly-revised version that takes into account the reviewers' comments.

Your paper hhas been reviewed with great care by several experts. One had a very large number of comments which are supplied on an annotated version of your submitted pdf. Please go through all of this. No more new experiments are needed but there are numerous contentious issues regarding data interpretation.

We cannot make any decision about publication until we have seen the revised manuscript and your response to the reviewers' comments. Your revised manuscript is also likely to be sent to reviewers for further evaluation.

Sincerely,

Christine Clayton

Associate Editor

PLOS Pathogens

Margaret Phillips

Section Editor

PLOS Pathogens

Kasturi Haldar

Editor-in-Chief

PLOS Pathogens

orcid.org/0000-0001-5065-158X

Michael Malim

Editor-in-Chief

PLOS Pathogens

orcid.org/0000-0002-7699-2064

I leave the metabolomes to the other reviewers but have myself looked at the transcriptomes. My comments are as follows:

Line 181 - why the lower alignment % of T. brucei reads - was it because the genome used for the alignment didn't include relevant VSGs?

Line 183: "to minimize artefacts from multigene families, only uniquely aligned reads were used for downstream analyses". You can't do this since it excludes many very important genes, e.g. glycerol kinase, aldolase.... For PGKs you can use the 3'-UTRs as well. Please also analyse the multi-gene families - or at least, those that are important. For known important families you can simply sum the reads that come from the different paralogues. If you remove all multipy aligned reads you also get completely distorted TPM values.

Transcriptomes must be compared using an appropriate tool designed for such analyses such as EdgeR or DeSeq2.

What are the relative division times of T. congolense and T. brucei in culture? Could this account for some differences in e.g. glucose consumption? Also transcriptomes obtained at peak parasitaemia are not comparable to those obtained in growing cells. To compensate for this I suggest that instead of thhe T. brucei transcriptome that was taken when the cells were too dense, you use some of the many publicly available datasets for bloodstream-form T. brucei that were not so dense - ideally, less than 2 million per ml. Cell densities of the parasites used for RNASeq must be stated, and the stage in the growth curve given. (Ideally this should be early-to-mid log phase and at least 2 divisions before the maximum density.)

Line 306 You cannot trust the gene copy numbers in the T. brucei 927 assembly because, due to the way in which is was generated, identical repeats are eliminated. See BMC Genomics volume 16, Article number: 1118 (2015) (Table S1) for some numbers. For example, 927 has 4 copies of glycerol kinase, and other strains (including field isolates) up to 9.

Table S1 -

1. Please include a raw data table (mapped reads for all genes, not just the unique reads) as a separate sheet.

2. Please round RPMs appropriately (e.g. to 1 decimal place) - makes it much easier to see differences.

Other comments:

1. Title and abstract should be re-worded -since it is misleading. Instead of "drug sensitivity" it should be "sensitivity to metabolic inhibitors" since most of the compounds used are not used as anti-trypanosomal drugs - or in some cases, not used as drugs at all. I agree with the reviewer who says that this part should be very much down-played. The most that you can say is that these results will help in selecting compounds that work on all species. The proportions of different species are quite variable across Africa; no African farmer is interested in having a drug that treats only T. congolense, so no company is going to try to market such a drug.

Line 141 - typo - "BSF T. congolense primarily expresses cytosolic PGK-C, rather than glycosomal PGK-B". Wrong way round, PGKC is glycosomal.

Reviewer's Responses to Questions

**Part I - Summary**

Reviewer #1: This manuscript describes extensive and detailed multi-omics approach to study metabolism of the bloodstream forms (BSF) of Trypanosoma congolense, in comparison with that of the bloodstream forms of T. brucei, which has been widely studied so far.

In contrast to T. brucei, the literature on metabolism in T. congolense is scarce. It was already known that BSF T. congolense consume and metabolize glucose in a different way compared BSF T. brucei and has developed a sort of T. brucei procyclic-like behavior. This comprehensive analysis provides important knowledge on this subject that will inform the path towards the development of new anti-trypanocidal compounds to fight against this causative agent of animal African trypanosomiasis, which is responsible for millions of livestock deaths annually.

This work also provides a detailed map of the carbon source usage, including all amino acids and lipids. For instance, BSF T. congolense prefers lipids and fatty acids, rather than de novo biosynthesis of these essential factors, which has been validated by the significantly less sensitivity of BSF T. congolense to inhibitors of fatty acid synthesis, compared to that of BSF T. brucei.

The work has been excellently performed mostly using expert metabolomics approaches and the manuscript has been well written.

I have only a few minor comments.

Reviewer #2: In this work, Steketee and co-workers make a comprehensive analysis of the metabolism of bloodstream forms of Trypanosoma congolense cultured in two conditions (in vivo and in vitro) and compare them with the much more known bloodstream forms of Trypanosoma brucei. The authors claim that this comparison could be useful for detecting in T. congolense “areas of the metabolism” suitable for intervention with drugs. The quantity of high quality information in this work and the excellent level of the analysis is fascinating. However, in the opinion of this reviewer, many aspects of the manuscript must be improved in order to become a digestible piece of information.

Reviewer #3: This manuscript is a substantial amount of work and is full of very useful information and data for the community in the coming years.

**Part II – Major Issues: Key Experiments Required for Acceptance**

Reviewer #1: No major issues

Reviewer #2: No additional experiments are required by this Revivewer.

Reviewer #3: There are several issues about metabolic identity. i.e. Table 2 what is dactyl and trihydroxy-fatty acids (C18:1) is not a realistic assignment. Fig 2B, unsure how the authors know that it is "12"-hydroxydodecanoic acid, as this is not a standard.

Fig 2 Does the total amount of choline, choline-phosphate, glycerol-P-choline observed in the media correlate with the amount of lyso-PC that has been degraded.

Fig 2: Why is there inosine 1 and 2 ?

Fig 2B Lyso PC(17:0), not of eukaryotic origin to any great extent, but this is not in Fig 2A??

Very concerned that the authors find some many aldehyde containing metabolite.

Fig 5: can the authors say anything about cytosine, as T. brucei do not take up it up, do you know if T.co does?

Fig 6 can the authors comment on deamination or decarboxylation of amino acids, other then glutamine/glutamate and Leucine, how about some of the aromatic amino acids Tryptophan?

There are a lack of ref for some of the comment in the amino cid metabolism section, i.e. serine and cya and trypanothione.

It is very interesting that fatty acid synthesis is not observed from Glc/Thr. Do we know if the GPI anchors in T.co still utilise Mys

Was lipoid acid observed in the analysis,as this is always de novo synthesed by kinetoplasids?

Did the authors observe any label from Glc or The into the mevalonate pathway, not necessarily sterol, but farsenyl and other isoprene products. As there are difference in the expression levels of these enzymes HMGCR, IDI, MDD etc?

Do the authors think it is worth looking at the SHAM Ec50 in the presence of higher glycerol concentration, as this in Tb is affected?

**Part III – Minor Issues: Editorial and Data Presentation Modifications**

Reviewer #1: 1- In lines 191-195, it is mentioned that some metabolic genes are not present in the T. congolense gene, but it would be informative to show the synthenic maps of these areas between T. brucei and T. congolense to illustrate the possible gene deletion in the T. congolense genome (supplementary data), and are metabolic genes absent in the T. brucei genome, compared to the T. congolense genome? For example, it is not mention in this list that MCP1 is missing in the T. congolense genome. This synteny-based analysis would be helpful to confirm the gene deletion.

2- In Figure 2B, the metabolite names are shown on the figure and in the figure legend. This is redundant.

3- In Figure 3, for convenience it would be helpful for non-specialists to include in Fig 3I the step numbers of Fig 3G. It would also be useful to indicate the FBPase step in Fig 3G, for instance by 3b (3a been phosphofructokinase).

- The figure legend mentions that "typically, the succinate shunt is only active in PCF T. brucei, with low levels of activity in BSF T. brucei". If low levels of activity of the succinate shunt have been described in BSF T. brucei, it is not ONLY active in PCF T. brucei. This should be rephrased.

- There is no glucose transporter in the glycosomal membrane since glucose is exchanged between the cytosolic and glycosomal compartment through a non-specific channels. Thus, the number 1 in Panel B has to be removed. In addition, step 12 should be removed since it PEP exchange between the cyotosolic and glycosomal compartments through the same non-specific channels. Thus, PEPCK corresponds to step 12, etc. until production of alanine and acetate.

4- In pages 13-14, there are some redundancy between lines 307-308 and 333-340, such as "to confirm that the elevated levels of succinate and malate seen in T. congolense spent medium samples originated from glucose ..." (lines 307-308) and "to determine whether the elevated succinate in supernatants originated from glucose catabolism, metabolite labelling was corrected …" (lines 333-334). This part can be shortened.

5- Lines 334-336 describes how the authors corrected the metabolite labelling in order to compare with the theoretical 1:1 (50%) ratio of natural glucose to 13C-U-D-glucose. Since, this correction is meant to get a 50% of 13C-U-D-glucose (instead of 43.1%), how >90% labelling in glycolytic metabolites up to pyruvate can be obtained, since it should not exceed 50%? Theoretically, these values should not exceed 50%. This, it seems that 100% labelling of glucose has been considered as a reference (instead of 50%) to analyse these data set. This is not correct. The authors should use as a reference 50% of labelling for glucose and correct the values of other metabolites accordingly.

6- Line 349, ref 37 should be included with ref 35.

7- The sentence lines 349-351 is confusing "similarly, no citric acid cycle intermediate isotopologues (e.g. citrate) were found when BSF T. congolense were incubated with 13C-U-D-glucose, although small amounts of 2-carbon labelled succinate and malate were observed (Fig 3H). Does it mean that 2-carbon labelled succinate and malate could be generated through the TCA cycle activity? Actually, in trypanosomes, 2-carbon labelling of succinate and malate (instead of 3-carbon labelled) is mainly due to the reversibility of the glycosomal succinate branch (from PEP to fumarate) and the symmetry of the fumarate molecule, which makes carbons 1 and 4 indistinguishable by fumarase. Indeed the reverse reaction catalysed by PEPCK (decarboxylation of malate to PEP), can randomly decarboxylates carbons 1 or 4 and thus remove 13C-carbons (50%), which is then replaced by a non-13C-enriched carbon through the forward PEPCK reaction. Thus, production of 2-carbon labelled succinate and malate can easily be explained by the reversion of the glycosomal succinate branch without invoking a possible TCA cycle activity.

8- Lines 389-390, in the presence of glucose as performed all along the manuscript, PPDK is not an enzyme of the succinate shunt since it is used to produce pyruvate from PEP. Consequently, this RNAi analysis is not relevant to the question asked. In addition, the low efficiency of the RNAi approach to knockdown PPDK (50%) does not allow a conclusion as to the role of PPDK. This experiment should be removed from the manuscript.

9- In the same line, RNAi-mediated knockdown of PEPCK is inconclusive for the same reasons, since 50% of the mRNA remains after induction. Instead of measuring the transcript levels, it would be more relevant to measure the PEPCK activity or effect of PEPCK partial knockdown on succinate production from metabolism of glucose. Anyway, this experiment should also be removed from the manuscript.

10- In figure 7, the ACS abbreviation has been used both for acetyl-CoA synthetase and acyl-CoA synthetase. To avoid confusion acetyl-CoA synthetase should be abbreviated AceCS as previously done. In line 1281, T. congolense should be in italics.

11- Line 414, remove the "s" from alternatives.

12- In line 415 and 618-619, the authors refer to Trindade et al. 2016 (ref 72) to support the T. brucei's use of b-oxidation for energy production. In fact, this work doesn't show it and the use of fatty acids for ATP production has not been demonstrated in T. brucei so far. Ref 72 showed that myristate (C14 FA), which is not myristyl-CoA (substrate for b-oxidation), can be reduced, hydrated and then reduced once more. However, production of C12 FA (or shorter FA), with the expected production of acetyl-CoA required for energy production, has not been observed. Actually, Trindade et al. did not conclude that BSF produce ATP from fatty acid catabolism. Thus, the author should not used ref 12 to support their hypothesis regarding the role of fatty acids in energy production of T. congolense.

13- In lines 418-419 is stated that "BSF T. brucei do not detectably express any ETC components with the exception of the reversed F1Fo-ATPase and alternative oxidase". Actually, one co-author of this manuscript published in Ref 92 evidences that complex I of the respiratory chain is expressed and functional in BSF T. brucei. These data should be taken into account and included in the discussion of the ETC components, including complex I.

14- Line 453, to help the reader it would be good to include in the text the abbreviation of hypoxanthine-guanine phosphoribosyltransferase and uracil phosphoribosyltransferase used in the corresponding figure.

15- Lines 601-602 (and Figure 7C), Since ACH has been mentioned in this section, it would be relevant to include data for the ASCT (TbTc_0236), which is also involved in acetate production from acetyl-CoA, but coupled with ATP production.

16- Line 683, "was" can be replaced by "were".

17- Line 732, what about Fumarate reductase (FRDg), which is the last step of this succinate-producing shunt?

18- Lines 784-788, the authors misunderstood the structure of the mitochondrial pyruvate carrier (MCP) composed of two small hydrophobic paralogous proteins, MPC1 and MPC2, which are essential and sufficient for the transport of pyruvate into mitochondria. Thus, MCP is composed of MCP1 plus MCP2, meaning that the absence of MCP1 in the T. congolense genome should be considered as a problem. The authors wrongly considered that T. brucei contains 2 MCP transporters (MCP1 and MCP2), while T. congolense a single one (MCP2). Consequently, they have to reconsider their analysis of this unexpected situation in T. congolense, since the production of acetate from glucose implies the existence of a functional mitochondrial pyruvate carrier.

19- In line 835, it has not been demonstrated that the putative T. brucei CL gene codes for a CL. Thus, it would be better to replace "this gene" by "this putative CL gene".

Reviewer #2: First, this reviewer, who is not native in English, feels uncomfortable by pointing the many mistakes in the use of this language. Some of them are pointed in the annotated pdf attached. Of course the Reviewer is aware that he/she can be wrong about some of the pointed mistakes but he/she is sure that the MS must be revised. About the style, the authors chose to produce a MS in which Results and Discussion are separated sections, however most of the discussion is made in the Results section. As the MS “touches” many different aspects of metabolism and makes many comparisons and links with the literature, this option is understandable, but makes the Discussion section highly repetitive, with little addition of information and/or ideas. I suggest to maintain the structure but shortening considerably the Discussion and limiting it to the aspects that are not discussed in Results. For example, the authors could privilege a more integrated analysis of the obtained data. This is just a suggestion and the authors should not consider this mandatory. Something else that the authors should reconsider is the claim that the article “highlights potential areas that are exploitable for pharmacological intervention”. In fact, from the amazing quantity of information offered to the reader there are little clues on new points of intervention with drugs. In the opinion of this reviewer this claim should be touched down.

Reviewer #3: There are a few grammatical and numerous puncuation things that need to be rectified.

Do not feel as though Fig 8 is required , could be supply fig, as the data is presented in Table 1.

In table 1 legend LCFA is define as an abbreviation, but not used in the table?

Line 712/713 22:4 to 22:4 , should be 22:5

PLOS authors have the option to publish the peer review history of their article (what does this mean?). If published, this will include your full peer review and any attached files.

Reviewer #1: **Yes: **Frédéric Bringaud

Reviewer #2: No

Reviewer #3: No
---

## [Decision Letter · Decision Letter 1]

21 Jun 2021

Dear Dr Steketee,

We are pleased to inform you that your manuscript 'Divergent metabolism between Trypanosoma congolense and Trypanosoma brucei results in differential sensitivity to metabolic inhibition' has been provisionally accepted for publication in PLOS Pathogens.

Best regards,

Christine Clayton

Associate Editor

PLOS Pathogens

Margaret Phillips

Section Editor

PLOS Pathogens

Kasturi Haldar

Editor-in-Chief

PLOS Pathogens

orcid.org/0000-0001-5065-158X

Michael Malim

Editor-in-Chief

PLOS Pathogens

orcid.org/0000-0002-7699-2064

Reviewer Comments (if any, and for reference):

Reviewer's Responses to Questions

**Part I - Summary**

Reviewer #1: The authors have responded appropriately to the quetsions and comments raised by this reviewers and apparently by the other reviewers as well.

Reviewer #2: In my opinion this is a very interesting manuscript and in its present version is suitable for publication. I am acknowledged for the extensive and detailed explanations given by the authors about my comments and critiques. The Tryps community undoubltly will benefit of the publication of such a large and complete set of data.

**Part II – Major Issues: Key Experiments Required for Acceptance**

Reviewer #1: (No Response)

Reviewer #2: (No Response)

**Part III – Minor Issues: Editorial and Data Presentation Modifications**

Reviewer #1: (No Response)

Reviewer #2: (No Response)

PLOS authors have the option to publish the peer review history of their article (what does this mean?). If published, this will include your full peer review and any attached files.

Reviewer #1: No

Reviewer #2: No

---

## [Editor Report · Acceptance letter]

20 Jul 2021

Dear Dr Steketee,

We are delighted to inform you that your manuscript, "Divergent metabolism between *Trypanosoma congolense* and *Trypanosoma brucei* results in differential sensitivity to metabolic inhibition," has been formally accepted for publication in PLOS Pathogens.

Best regards,

Kasturi Haldar

Editor-in-Chief

PLOS Pathogens

orcid.org/0000-0001-5065-158X

Michael Malim

Editor-in-Chief

PLOS Pathogens

orcid.org/0000-0002-7699-2064